# Algorithmic Heap Layout Manipulation in the Linux Kernel

Max J. Ufer
*Fraunhofer FKIE*
*max.jens.ufer@fkie.fraunhofer.de*

Daniel Baier
*Fraunhofer FKIE*
*daniel.baier@fkie.fraunhofer.de*

## Abstract

To evaluate the severity of a security vulnerability a security researcher usually tries to prove its exploitability by writing an actual exploit. In the case of buffer overflows on the heap, a necessary part of this is manipulating the heap layout in a way that creates an exploitable state, usually by placing a vulnerable object adjacent to a target object. This requires manual effort and extensive knowledge of the target. With a target as complex as the Linux kernel, this problem becomes highly non-trivial. At the current time, there has been little research in terms of employing algorithmic solutions for this. In this work, we present Kernel-SIEVE, a framework for evaluating heap layout manipulation algorithms that target the SLAB/SLUB allocator in the Linux kernel. Inspired by previous work that targets user-space allocators [33–35] it provides an interface for triggering allocations/deallocations in the kernel and contains a feedback loop that returns the resulting distance of two target objects. With this, we create the (to our knowledge) first performance benchmarks for heap layout manipulation algorithms in the Linux kernel. We present and evaluate two algorithms: A pseudo-random search, whose performance serves as a baseline, and KEvoHeap, a genetic algorithm based on Heelan's EvoHeap [33, 35]. We show that KEvoHeap is successful at creating the desired heap layout in all test cases and also surpasses the user-space performance benchmarks of EvoHeap. Finally, we discuss the challenges of applying these kinds of algorithms in real-world scenarios and weigh different possible approaches to tackle the problems that arise. Our research results are publicly available on GitHub [43].

## 1 Introduction

From a normal user's perspective, Linux seems to fall far behind other operating systems. In May of 2021, only 2.38% of desktop computers used Linux as an operating system, in contrast to the overwhelming lead of Windows with 73.54% [6]. One might think that it is only used by programmers or other computer enthusiasts. But if you take a look behind the curtain, it becomes apparent what a dominant role Linux plays for almost everyone. While not being dominant on desktops, according to W³Tech 75.6% of the top 10 million websites use Unix-based operating systems, of which 56.1% are Linux systems [55]. In 2015, of the top 25 most popular websites according to Alexa [5], only two did not use Linux. Coincidentally, these two websites were live.com and bing.com, which are both owned by Microsoft [54]. In addition to the web, Linux is also very prevalent on mobile devices. Android, which uses a modified Linux kernel, has by far the biggest market share on smartphones with 72.72% [10] and is also the overall leading operating system across all devices [11]. With such a large and distributed usage naturally comes the attention of malicious actors such as black-hat hackers or even Nation-State actors who try to find and exploit vulnerabilities in the operating system itself. These kinds of Linux kernel vulnerabilities are especially dangerous as they can lead to full control over the device with root access on a variety of different kinds of devices like phones, desktop computers, servers, and today even IOT devices or cars. They are also very valuable: Zerodium pays up to $50.000 for an exploitable local privilege escalation bug [14].

As the kernel is an incredibly complex piece of code, consisting of over 27 million lines of code [17], security-related bugs are being reported regularly. Because of the sheer number of bugs being reported every day, they can not all be addressed immediately. Instead, they have to be prioritized [53]. One factor when evaluating a security-related bug is if it is actually exploitable. For this, the analyst has to create a working exploit using the presented vulnerability to prove its exploitability. As this can be a very difficult and time-consuming process (especially when it comes to kernel exploitation), ways to automate parts of this task were researched.

While some parts in exploit development like shellcode creation have been successfully automatized, many tasks still require manual effort by the exploit developer. One challenge which one often faces when dealing with memory corruption vulnerabilities regarding the heap is interacting with the target in a way that brings the heap into an exploitable state. This is often referred to as Heap Layout Manipulation [34] or Heap Feng Shui [50]. While not much work has been done regarding automating this process, it gained some attention in the last years in the context of fully automatic exploit generation [22, 23, 33–35, 56, 61].

In this paper, we will evaluate the performance of algorithms designed to solve the heap layout manipulation algorithm in the kernel. For this, we created Kernel-SIEVE, a framework inspired by previous work from Sean Heelan [33–35] to create heap layout manipulations in the Linux

kernel and provide an interface for algorithms to solve them. With this, we evaluated two algorithms: A pseudo-random search that serves as a benchmark for further experiments, and KEvoHeap, a genetic algorithm based on Heelan's EvoHeap algorithms. In the end, we provide an outlook on the challenges that have to be overcome to apply these kinds of algorithms to real-world vulnerabilities and possible ways to overcome them.

The solution we present is aimed towards a security researcher that found a heap-overflow or underflow bug in the kernel and wants to prove its exploitability. In the process of finding a solution for the heap layout manipulation problem we rely on tools like bpftrace [48] that require root access, but these privileges are not needed to execute the final solution (cf. 11).

In summary, this paper makes the following contributions:

- We present Kernel-SIEVE, a framework for evaluating heap layout manipulation algorithms that target the SLAB/SLUB allocator in the Linux kernel.

- We propose and evaluate two algorithms for creating desired kernel heap layouts: A pseudo-random search, whose performance serves as a baseline, and KEvoHeap, a genetic algorithm based on Heelan's EvoHeap.

- We provide scripts to visualize candidate solutions in an animated fashion.

- We provide a vulnerable kernel module that serves as a case study containing a heap buffer overflow vulnerability to demonstrate these types of algorithms in real-world scenarios and their application in the exploit development process.

Following our belief in open research, we provide everything as open source on GitHub [43].

## 2  Related Work

The task of automatically adjusting heap layouts into an exploitable state is a sub-task of automatic exploit generation. This is a relatively novel field of research, with most publications dating back only 5-10 years. In this section, we will give an overview of the current state of research that is either relevant to this work or give otherwise beneficial context for the broader field of exploit automation.

### 2.1  Automation of Exploitation Sub-tasks

In 2018 Wu et al. proposed FUZE [59], a framework that aims to facilitate the exploitation of use-after-free vulnerabilities in the Linux kernel. It uses a combination of kernel fuzzing and symbolic execution to identify system calls that can be useful for exploiting a given kernel use-after-free vulnerability. These primitives are also evaluated according to their

usefulness for actual exploitation. Building upon this they presented KEPLER [58], a framework that takes a control-flow hijacking primitive in the kernel and generates a bootstrapping payload for kernel-ROP based shellcodes. Continuing their work, Chen et al. proposed SLAKE [23], a system that uses a combination of static and dynamic analysis to identify allocation and deallocation primitives in the Linux kernel. It creates a database of kernel objects useful for exploitation and the system calls which cause their (de-)allocation. With this, it can also try to adjust the SLAB layout to allow exploitation of a given Use-After-Free, Double Free, or Out-Of-Bounds-Write vulnerability. Their most recent publication in this line of work is KOOBE [22], a framework that aims to assist a researcher while analyzing out-of-bounds write vulnerabilities in the kernel. Given a PoC, it evaluates the vulnerability's capabilities and checks if they are sufficient for successful exploitation. If yes, it tries to generate a full exploit, incorporating existing Heap Feng Shui techniques. If not, it uses a novel kind of fuzzing to explore new capabilities of the given vulnerability.

In 2018 Heelan et al. published a paper called "Automatic heap layout manipulation for exploitation" which they claimed to be the first one to address the topic of automatic heap layout manipulation [34]. In their paper they presented two evaluation frameworks for heap layout manipulation algorithms: SIEVE, a framework for creating synthetic challenges on different allocator implementations, and SHRIKE, a heap layout manipulation system for the PHP interpreter. In addition to providing an interface for algorithmic solutions, it also solves some real-world problems like extracting primitives for heap layout manipulation. They used the pseudo-random search to automatically create heap layouts and provide a benchmarking baseline for future work.

One of the most recent works regarding the automatic adjustment of heap layouts is MAZE [56], a framework created by Wang et al. It models the heap and the available interactions with the allocator as a Linear Diophantine Equation and solves it deterministically to find an interaction sequence that results in the desired heap state. It also can discover heap manipulation primitives through static analysis.

### 2.2  Automatic Exploit Generation

The automatic exploit generation challenge can be defined in two different ways. The "easy" formulation which is used most of the time, where the system gets a vulnerability, e.g. in the form of a PoC program, and should output a full exploit that (usually) spawns a shell, or the "hard" formulation, in which the system also has to find the vulnerability by itself. AEG [19] claims to be the first system that solves the hard version of this challenge. It uses source code analysis and symbolic execution to identify vulnerabilities and then generates a payload under consideration of input constraints. AEG only targets stack overflow and format string vulnerabilities.

In 2020 Sean Heelan published his Ph.D. thesis about "Greybox Automatic Exploit Generation for Heap Overflows in Language Interpreters" [33]. He presented a greybox approach for generating exploits for existing heap overflow vulnerabilities without relying on symbolic execution or other whitebox methods. It builds upon GOLLUM [35], a previous publication of his which claimed to be the first framework for automatically generating heap overflow exploits in language interpreters. They employ a modular approach, using the previously presented SHRIKE [34] system for solving the heap layout manipulation problem and a new approach for identifying new exploit primitives from tests. Their system relies on multiple assumptions made about the target, e.g. that a break for ASLR is available and control-flow integrity protection is not deployed.

In 2018 Eckert, Moritz, et al. proposed HEAPHOPPER [26], an automated approach, based on model checking and symbolic execution, to analyze the exploitability of heap implementations in the presence of memory corruption. Using HEAPHOPPER, they were able to perform systematic analysis of different, widely used heap implementations, finding surprising weaknesses in them.

HAEPG [61] is an automatic exploit generation framework proposed by Zhao et al. in 2020. It utilizes symbolic execution to exploit heap-based vulnerabilities using provided exploit templates. It takes a crashing input as an input and outputs a complete exploit, which e.g. spawns a shell. In contrast to other works, HAEPG can bypass NX [45] and Full RELRO [49].

## 2.3   Kernel Exploitation

One of the most referenced resources for kernel exploitation is the book "A Guide to Kernel Exploitation: Attacking the Core" [46] by Enrico Perla and Massimiliano Oldani. It provides a broad overview of kernel exploitation techniques and their application on Mac OS X, Windows, and operating systems of the UNIX family. In terms of Linux, it also gives an introduction to the inner workings of the SLAB/SLUB allocator. In the article "Linux kernel heap feng shui in 2022" [44], the authors provide an overview of the Linux kernel slab allocator implementation and its exploitation challenges associated with kernel heap-related vulnerabilities. Major changes in the Linux kernel that affect the exploitability of heap-related vulnerabilities and their exploitation strategies are discussed. Besides this, there is a large number of articles and blog posts available going into detail on the exploitation of Linux kernel vulnerabilities [51,62] and the exploitation of variants like the Linux kernel fork used in Android [18,52].

In 2022 Zeng, Kyle, et al. "Playing for K (H) eaps: Understanding and Improving Linux Kernel Exploit Reliability." [60] the authors provide a systematic study of the kernel heap exploit reliability problem. Through this a generic heap exploit model is presented. This model explains the process of kernel heap exploitation, spanning from the moment that an exploit starts to interact with a vulnerable system to the moment that the exploit successfully triggers an attacker-controlled payload.

## 3   Exploiting Heap Overflows

While allocating on the stack may be sufficient for variables of static size and that are only used in the scope of a function, it is not suited for allocations of dynamic size or allocations that should persist after the function returns [47]. For these cases, space can be allocated on the heap.

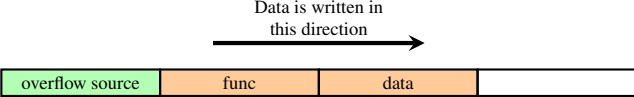

Figure 1: Heap-layout for corrupting an example structure containing a function pointer and some data.

The heap is a memory region that is controlled by a heap allocator. Its behavior is not defined in the ANSI C standard [16] but depends on its implementation. The most common interface used for allocating memory in user space is the `malloc` function. In the Linux kernel its counterpart `kmalloc` is used. When called, it allocates a contiguous block of memory of the requested size and returns its address. More details on the internals of kernel heap allocation will follow in the next section.

```
char* copy_to_buffer_heap(char *input){
    char buffer = kmalloc(16 * sizeof(char,
            GFP_KERNEL);
    strcpy(buffer, input);
    return buffer;
}
```

Above you can see a simple function that allocates a 16-byte buffer on the heap, copies the string that `input` points to to that buffer, and returns its address. Here we have a potential buffer overflow. However, now we can not overwrite the return address like we would in case of a stack overflow, as we overflow into the heap. On its own, this function is not exploitable. However, this vulnerability can be used to overwrite data in an adjacent allocated object. The listing below illustrates how such a target may look like.

```
typedef struct target {
    void* (*func)();
    char *data;
} target_t;
```

This struct contains a function pointer as its first element. If we can force the program (in this case, the kernel) to

allocate this structure on the heap and also manipulate the heap allocator in a way that this allocation will be placed directly next to our overflow source, as illustrated in Figure 1, we could overwrite the function pointer and also the data pointer in the adjacent object. Now execution flow can be redirected after triggering a call of `func`.

The above-described exploitation approach is only one of many. Other ways include meta-data corruption or overwriting useful fields in different kinds of structs which may not be function pointers but could lead to different kinds of primitives like arbitrary read-write. What all these approaches have in common is the need to place a target object next to a vulnerable object. The process of creating an exploitable heap state is usually called Heap Feng Shui [50], heap grooming [29] or heap layout manipulation. Finding a solution for this problem is not trivial and becomes even harder in the kernel because of multiple factors:

- **Indirect allocator access:** While the heap allocator has a direct interface, from an exploit writer's perspective we can not directly access it. The standard way to interact with the kernel is via system calls. So what we have to do is use those system calls that will trigger allocations which will be beneficial to our goal. Unfortunately, as the kernel is an incredibly complex piece of software, triggering a system call might make multiple allocations of the same or different objects, which cause side effects to the heap, making it harder to achieve a useful heap layout. These side effects will be referred to as *noise* for the remainder of this work. For example, let's assume we have found primitives which can cause the following allocations: We have one primitive which solely allocates the overflow source, and a second one, which allocates the target, but always first allocates another object. Additionally, we are able to trigger a deallocation of the overflow source. If we simply call both allocation primitives sequentially, the additional allocation of the second primitive will be placed between the overflow source and the target (we assume that for successful exploitation they have to be directly adjacent)[1]. To solve this problem, we can use the primitives we have to carefully craft a heap layout which still results in the overflow source and the target being adjacent. There are multiple solutions for this example, but one may look like this: We first allocate the overflow source two times, and then trigger a free of the first one. This creates a "hole" in memory before the overflow source. Now we trigger the allocation of the target. The additional object, which is created by the primitive, will now be placed in said hole, while the target results adjacent to the overflow source. Figure 2 illustrates this process.

---

[1]In this scenario we assume that we already have an unfragmented heap state with a linear free list. In Section 5 we will show that this can be achieved

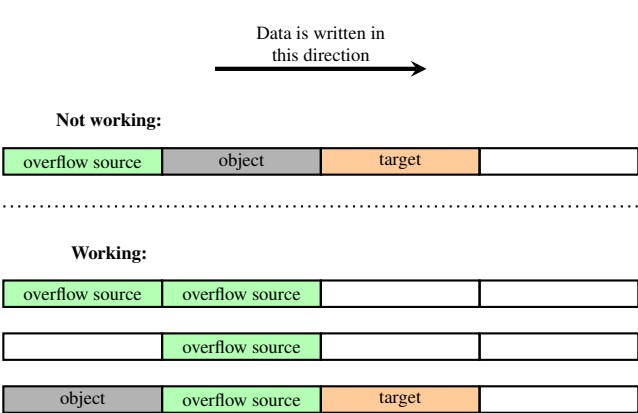

Figure 2: Example illustration for how sub-optimal primitives can still be used to create a desirable heap layout.

- **Unknown initial heap state:** When we start our exploit and thus our manipulation of the heap layout, we do not know the current heap state. When you execute a user space program twice, we can at least expect to get the same heap layout twice (assuming the use of a deterministic heap allocator). In the kernel, it is even worse due to multiple factors. First, when we start our exploit, the kernel is *already running*. Because of the massive complexity of the kernel and many background threads/processes which also influence the kernel heap, this makes it impossible to guess the initial heap state. Secondly, executing the exploit multiple times can also alter the heap in a way that the first execution influences the second one. As we will see in the next section, the kernel heap allocator keeps free objects in a free list. For example, if the exploit triggers multiple allocations and frees, this free list will be in a different state when we execute the exploit the second time. As a result, the allocator will behave differently. In Section 5 we will show how to solve these problems methodically.

- **Indeterministic behavior of the heap allocator:** The Linux kernel contains a configuration option called `CONFIG_SLAB_FREELIST_RANDOM`. When enabled, this will randomize a cache's free list on initialization. This option is out of scope for this work, as it is disabled in the Linux default configuration and the approach we are taking to solve the heap layout manipulation problem requires a workaround for the free list randomization. Adding a free list de-randomizer to the algorithm would be a great starting point for further research.

## 4  Memory allocation in the Linux kernel

The Linux kernel is the core of many modern operating systems. This includes Linux-based operating systems like

---

methodically.

Ubuntu [21] and Debian [15], but also e.g. the Android operating system, as they use a modified Linux kernel at its core [31]. While it works mostly in the background, a user can interact with it via system calls. In this section, we will describe how memory is managed in the Linux kernel and which heap allocator implementations are available.

Directly on the physical memory sits the "Buddy Allocator" [32, 36] that maps physical memory pages into virtual memory. The different user space allocators receive pages directly from the buddy allocator and implement different allocation strategies on these. In kernel space, on top of the buddy allocator sits the "Slab Layer" that exposes the general-purpose allocation interface `kmalloc`. There are different implementations for the Slab Layer, but almost all modern distributions use the SLUB allocator, which is the modern default.

## 4.1 Slab-Allocation

The slab allocator is Linux's general-purpose allocator and sits on top of the buddy allocator [25, 32]. Slab allocation was first used in OpenSolaris, and the Linux version of it is heavily based on theirs [25, 39, 46]. The main purpose of the slab allocator is to provide a way of allocating small objects in an efficient way and cache commonly used objects to improve allocation, initialization, and destroy timings. Over the last 30 years, slab allocation in Linux has evolved and changed drastically. Today there are three different allocators between which the user can decide before building the kernel: SLAB, SLUB, and SLOB.

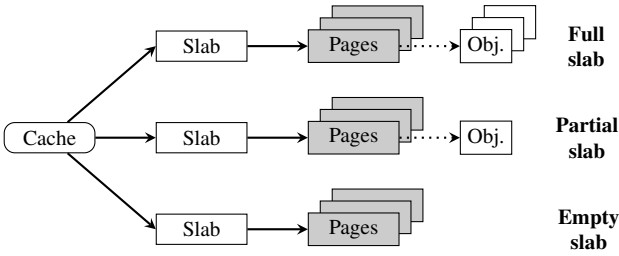

Figure 3: The structure of caches and slabs.

Before we start discussing slab allocation, we need to differentiate between a few somewhat ambiguous terms:

- slab allocation: General memory management strategy

- slab: Contiguous physical memory pages, which can store data associated with objects

- SLAB, SLUB, SLOB: Different slab allocator implementations

One of the main data structures of a slab allocator is the cache. A cache in terms of slab allocation manages memory for a specific object type. One cache consists of multiple slabs, which themselves are blocks of contiguous physical pages of memory. The pages managed by a slab are cut into equal chunks of the size of the target object. Figure 3 illustrates this structure.

Internally, there are three classes of slabs in a cache: `slabs_full` (containing all slabs without free chunks), `slabs_partial` (containing slabs with free and non-free chunks), and `slabs_free` (containing only free chunks). Information about the different caches can be retrieved via `/proc/slabinfo`, as shown in Figure 4. For the sake of simplicity, we will only focus on the first five numbers that `/proc/slabinfo` gives for each cache, as these are also the most relevant for the later sections. The figure below shows an example output for the `task_struct` cache. The `task_struct` structure is the kernel representation of a user space process [41]. The first two numbers show how many active objects are held in the cache and how many are available in total. In this example, four more `task_struct` objects can be allocated, before the allocator has to create a new slab. The third number gives the size of the chunks which are available from that cache, so in this case 7872, which equals the size of one `task_struct`. The fourth number tells how many objects fit into one slab, and the fifth tells how many pages one slab consists of. The relation of numbers three to five is obvious: The total space available in the cache for `task_struct` is:

$$8 \cdot PAGE\_SIZE = 8 \cdot 4096 = 32768$$
$$\lfloor \frac{32768}{sizeof(task\_struct)} \rfloor = \lfloor \frac{32768}{7872} \rfloor = 4$$

So, if four more `task_struct` objects were allocated, there would be no partial or empty slabs left, so new slabs need to be allocated for further `task_struct` allocations.

Generally speaking, there are two types of caches: Caches of commonly used objects and sized caches. The first kind is particularly useful as the kernel has many structures which are allocated and deallocated many times. One example would be the previously mentioned `task_struct` structure. By keeping these objects in caches, allocation and deallocation times can be reduced by leaving an object in its initialized state when it is freed. When this object is allocated again, there is no need to initialize it again. The sized caches are not reserved for dedicated objects, but keep objects of certain sizes. These sizes are all powers of two. When an allocation is requested via `kmalloc` (the kernel's general allocation interface), a chunk from the next best fitting cache is returned. For example, when we try to allocate 33 bytes, we will actually get a 64 bytes chunk [25, 32]. The slab layer can also be circumvented by using the `vmalloc` interface. `vmalloc` accesses the buddy allocator directly and allocates memory that is virtually contiguous but can be physically scattered. As it comes with additional overhead and is slower than `kmalloc`, its usage is discouraged [24].

Nowadays the SLUB allocator is used, which replaced the original SLAB allocator as the modern default in Linux. A description of its implementation can be found in appendix A.

From an exploit writer's perspective, SLUB opens up new perspectives to heap exploitation:

- As the free list is stored as a linked list in the free chunks, this enables possibilities for metadata-corruption via overflowing into a free block and overwriting the pointer to the next object.

- As the slab pages are now only packed with objects, overflowing between page frame borders [46] could under certain circumstances be easier, as no metadata is corrupted in the process.

- One interesting property of SLUB is that it can combine slabs that contain different objects of the same size. This can open up new attack vectors for heap overflows, as we are less restricted in the kind of target we overflow into.

## 5 Kernel-SIEVE: Evaluating HLM Algorithms in the Kernel

The main goal of this work is to evaluate the usefulness of algorithmic solutions for the heap layout manipulation problem in the Linux kernel. For this, we developed a framework called **Kernel-SIEVE**, which enables us to create artificial heap layout manipulation challenges and provides an API for candidate algorithms to solve these challenges. This framework is inspired by SIEVE, the framework Heelan proposed for evaluating heap layout manipulation algorithms on different allocators in user space [33]. The base challenge this framework provides is to place two designated objects at a certain distance in the kernel heap. These objects represent an overflow source and a target. In this section, we will present the aforementioned framework and go into detail on the architecture and design decisions we made due to the challenges that arise when working in kernel space.

The architecture of Kernel-SIEVE is illustrated in Figure 5. It consists of two components:

- **Kernel Module:** The kernel module is Kernel-SIEVE's way of interacting with the kernel heap. It can be used to trigger the standard memory allocation operations `kmalloc`, `kfree`, `kcalloc`, and `krealloc`. It can be controlled via the `ioctl` system call. The kind of operation to perform is selected via the `request` parameter. Additional information about the allocation/deallocation to be performed are given via the struct `slab_params`:

```
struct slab_params {
    size_t size;
    id_t ID;
    size_t nmemb;
    id_t oldID;
    addr_t addr;
};
```

The members of this struct have the following purposes:

- `size`: The size of the allocation to make.
- `ID`: An ID to associate the allocation with, for later reference.
- `nmemb`: Only relevant for `kcalloc`: The number of elements to allocate.
- `oldID`: Only relevant for `krealloc`: The ID of the allocation that should be reallocated.
- `addr`: The resulting address of the requested allocation. This will be filled by the kernel module and used by the client to get information about the heap state.

- **Client:** The client is a program that performs interactions with the kernel module and provides an API to candidate algorithms. The main requirement when developing the client was that it must not have any unforeseen side effects to the kernel heap, which would falsify the results of candidate solutions. Modern programming languages like Python proved not to be viable options, as the runtime communicates with the kernel in unpredictable ways for garbage collection, forking, etc. Because of this, we chose to implement the client in pure C. This gave us direct control over all interactions the

```
$ cat /proc/slabinfo
# name <active_objs> <num_objs> <objsize> <objperslab> <pagesperslab> : tunables \
        <limit> <batchcount> <sharedfactor> : slabdata <active_slabs> <num_slabs> \
        <sharedavail>
[...]
task_struct          836     840    7872    4    8 : tunables   0    0    0 : slabdata   210 \
        210      0
[...]
```

Figure 4: Example output of `/proc/slabinfo`.

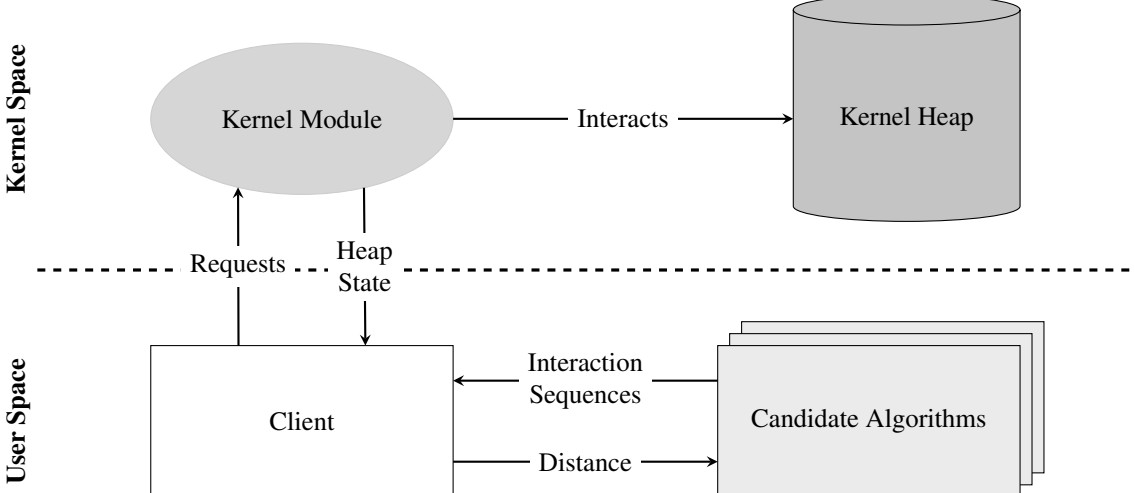

Figure 5: Architecture of Kernel-SIEVE.

client performs with the kernel. An algorithm can provide the client candidate solutions via files, where each file represents one candidate solution. This indirect form of communication comes with the advantage, that the programmer is free in his choice of language in which he wishes to implement the candidate algorithm in, as implementing complex algorithms in pure C can be quite exhausting. The client and the algorithm are executed in a loop via a script. The algorithm places his candidates in a dedicated directory. The client executes the candidate solutions and writes the resulting distances into a separate directory. By separating the execution of the algorithm from the client we prevent the algorithm from having side effects to the kernel heap from background processes during the execution of the client.

The files, that the algorithm has to create for the client are made of directives that are almost identical to the ones Heelan used for communicating with his SIEVE driver:

1. `<kmalloc size ID>`
2. `<kcalloc nmemb size ID>`
3. `<kfree ID>`
4. `<krealloc oldID size ID>`
5. `<fst size>`
6. `<snd size>`

The first four directives directly correspond to the standard allocation/deallocation interface of the kernel which was already described in Section 4.1. All allocations are assigned an ID, which is necessary to reference allocations for frees. The last two directives are used to allocate the two objects that eventually should be placed at a certain distance. The client takes a file consisting of those directives, translates them to parameters for the kernel module, and executes them sequentially. Finally, it calculates `addr(fst) - addr(snd)` and writes it to a result file.

While implementing this framework in kernel space, we faced certain problems that appear due to the nature of the kernel:

- **Unknown heap state:** One problem everyone faces in kernel heap exploitation is that you do not know the initial state of the heap when starting your exploit. This problem was already outlined in Section 3. Luckily, there is a methodical way around this called *heap defragmentation* [23, 46]. As stated in Section 4.1, we can use the `/proc/slabinfo` utility to collect information about the current state of the cache we target. Most importantly, it tells us how many active objects are in the target cache, and how many objects are available in it in total. When all available objects in the target cache are exhausted, the slab allocator has to map a new page frame into memory and create a new slab, which will have no active objects in it. So, if we allocate enough objects (specifically at least `num_objs - active_objs`), we can fill up all partial slabs and force the allocator to allocate a new one. If we do this every time before executing a candidate, we can create a predictable heap layout.

- **Unstable results and reproducibility:** When trying to solve the heap layout manipulation problem algorithmically, it is essential that we can rely on the results of our

candidates to be correct and reproducible, so that if we execute the same candidate twice, we will get the same result twice. Unfortunately, this is easier said than done when it comes to the kernel. The kernel runs many different threads in parallel, which all access the same heap. This means that there is always the chance that another thread performs operations on the same cache that we target just when we execute our candidate, falsifying the result. This is not the only problem: To not eventually run out of memory, the client frees all allocations we made again, clearing the slab. Due to the execution of the candidate, we now have a free slab with a reordered free list. This is a huge problem: The result of a candidate now depends on the previously executed candidates! Our way around this is to simply create our own cache, on which we now run our experiments. This cache will behave the same way as any other cache would, but will not be subject to random side effects by the kernel. There is a simple interface to manage caches [27]:

```
// "include/linux/slab.h"
struct kmem_cache *kmem_cache_create();
void kmem_cache_destroy();
void *kmem_cache_alloc();
void kmem_cache_free();
```

The kernel module has two modes of operation: It can either be configured to use the default `kmalloc` interface or to implement our "custom cache" strategy. Here, we create a dedicated cache for each candidate, which is destroyed after execution. Algorithms should use the "custom cache" mode, as this prevents the aforementioned instability problems. To be absolutely sure about the produced solution, you can switch into the "real" mode and check the solution with the default allocation interface. In our experiments, the results from the "real" mode did not differ from the "custom cache" mode, if no other candidates were executed before.

This framework enables us to create arbitrary challenges for candidate algorithms just like in Heelan's SIEVE. While the available directives allow very direct access to the allocator, the difficulty of the challenge can be adjusted in the implementation of the algorithm, for example by only allowing the algorithm to choose between certain *combinations* of directives to simulate the "indirect allocator access" problem which was explained in Section 3. The code is available on GitHub [43].

## 6 Candidate Algorithms

In this section, we will describe the algorithms that we chose to implement and evaluate. To create a baseline to compare other algorithms against, we first implemented the same

pseudo-random search that Heelan also used as a baseline [33]. This serves the additional purpose to compare the difficulty of the problem in kernel space to the different user space allocators Heelan evaluated. The second algorithm is *KEvoHeap*, a genetic algorithm that is a modified version of Heelan's EvoHeap algorithm. We made some modifications to adjust it to the special characteristics of the slab allocator. In the following sections, we will explain both algorithms in detail. A comprehension of Heelans work can be found in in appendix B.

### 6.1 Pseudo-Random Search

The algorithm we use to create a baseline for comparison is a pseudo-random search similar to the one used in Heelan's work [33, 34]. It is outlined in Algorithm 6. A description of the algorithm and its adjustments can be found in the appendix C.

### 6.2 KEvoHeap

The previously described algorithm mostly serves as a baseline, as it does not implement any real strategy. As described before, we consider the heap layout manipulation problem as an optimization problem regarding the distance of the two target objects. There are many approaches for numerically solving optimization problems. One of them is genetic algorithms. Genetic algorithms are a kind of evolutionary algorithms, which itself is a class of optimization algorithms that make use of basic evolutionary principles [20]. They start with an initial set of candidate solutions (called *population*), in which each is assigned a fitness value that represents the quality of the solution. The population iteratively passes multiple iterations (*generations*) in which first the *offspring* is generated from the population through mutation and/or crossover operations. The offspring then gets evaluated and assigned a fitness value each. From the evaluated offspring the new population is selected based on some principle aimed towards minimizing/maximizing the fitness value. [20]. In the following part, we will describe KEvoHeap, a genetic algorithm based on Heelan's EvoHeap [33] for solving the heap layout manipulation problem in the Linux kernel.

While KEvoHeap is based on Heelan's EvoHeap, we made some adjustments to it to better fit it to the SLAB/SLUB allocator. Also, the structure is different, as our Kernel-SIEVE framework has a different execution loop than SIEVE. Here we will explain the algorithm and point out similarities and differences to Heelan's EvoHeap. Algorithm 1 shows the main routine of KEvoHeap.

Usually, a genetic algorithm consists of a main loop that resembles the generational cycle of creating offspring, evaluation and selection. In Section 5 we explained that we decided to separate the execution of the client from the execution of the algorithm to get rid of side effects to the kernel heap. In-

**Input:** *target*, *μ*, *λ*, *mxpb*, *cxpb*
**Output:** A winning individual or nothing

**1 Function** EvoStep(*target*, *μ*, *λ*, *mxpb*, *cxpb*):
**2**   *pop* ← ReadPopulation()
**3**   *dist* ← ReadDistances()
**4**   **if** len(*pop* = 0) **then**
**5**     *pop* ← InitPopulation(*μ* + *λ*)
**6**     WritePopulation(*pop*)
**7**     **return**
**8**   **else**
**9**     *fit* ← Evaluate(*pop*, *dist*)
**10**     **for** *i* ← 0 **to** len(*pop*) − 1 **do**
**11**       **if** *fit*[*i*] = abs(*target*) **then**
**12**         **return** *pop*[*i*]
**13**     **end**
**14**     *survivors* ← Select(*pop*, *fit*, *μ*)
**15**     *offspring* ← GetChildren(*pop*, *λ*, *mxpb*, *cxpb*)
**16**     *pop* ← *survivors* + *offspring*
**17**     WritePopulation(*pop*)
**18**     **return**

**Algorithm 1:** The main routine of KEvoHeap.

**1 Function** GetChildren(*pop*, *λ*, *mxpb*, *cxpb*):
**2**   *children* ← []
**3**   **while** *λ* > 0 **do**
**4**     *parentA* ← *pop*[Random(0, len(*pop*))]
**5**     *r* ← Random(0, 1)
**6**     **if** *r* < *mxpb* **then**
**7**       *new* ← Mutate(*parentA*)
**8**       *children*.append(*new*)
**9**       *λ* ← *λ* − 1
**10**     **else if** *r* < *mxpb* + *cxpb* **then**
**11**       *parentB* ← *pop*[Random(0, len(*pop*))]
**12**       *newA*, *newB* ← Crossover(*parentA*, *parentB*)
**13**       *children*.append(*newA*)
**14**       *children*.append(*newB*)
**15**       *λ* ← *λ* − 2
**16**     **else**
**17**       *children*.append(*parentA*)
**18**       *λ* ← *λ* − 1
**19**   **end**
**20**   **return** *children*

**Algorithm 2:** Method to generate offspring for the next generation.

stead, the client and the algorithm are run alternating with a runner script. So, although there is no main loop, it exists implicitly. When the routine starts, the algorithm first reads the existing population and the resulting distances of the target objects from the dedicated directories (lines 2 and 3). If there is no existing population, that means we are in the first iteration and have to generate a new population and write it (lines 5 and 6). If there is, that means we are in the main cycle. First, we evaluate the population (line 9). Then we check if we found a solution, and if yes, return it (line 12). If not, we first select the survivors of the current generation (line 14). Then, we generate the offspring from the original population and write the new population, consisting of the survivors of the previous generation and the newly generated offspring. The GetChildren method is listed in Algorithm 18.

GetChildren creates *λ* children from *pop*. In each iteration it first selects a random individual from *pop* as a parent (line 4). Then it decides based on the mutation probability *mxpb* and the crossover probability *cxpb* whether to mutate the parent (line 7), perform a crossover with a random different individual (line 12) or to simply keep it as it is (line 17). As you can see the method is identical to the one used in EvoHeap with the small difference that we use both offspring created from the crossover in 12, while in EvoHeap only one is used (cf. B.1).

A detailed explanation how the algorithm represents its individuals and how each of the genetic operators was imple-

mented can be found in appendix D.

## 7   Evaluation

In this section, we will lay out how we evaluated the previously described algorithms. First, we describe a set of challenges that we designed to test the effectiveness of the different approaches. Then we show how the algorithms performed on the challenges and compare them.

### 7.1   Synthetic Benchmarks

To evaluate the algorithms we created a set of challenges for the algorithms to solve. We chose a similar design for the challenges as Heelan did [33], so we can compare the results between the different targets. In general, there are two kinds of challenges: Natural allocation order and reverse allocation order. In both challenges, the target object mimicking the overflow source has to be allocated first before the allocation of the target object. In the "natural" challenge, a normal buffer overflow situation is simulated, where the target allocation must follow the overflow source. In the "reverse" challenge, we simulate an *underflow* instead. Recall that the underflow source still has to be allocated first, so already in the simplest

of cases where the algorithm has direct control about the allocation of the target objects it has to find a slightly more complicated solution. An example solution is shown in Figure 6. Here we first allocate a placeholder object, followed by the allocation of the underflow source. Then, we free the placeholder object, which places the memory slot in front of the underflow source to the beginning of the free list of the slab. Lastly, we allocate the target object, placing it in front of the underflow source. The algorithms will have four kinds

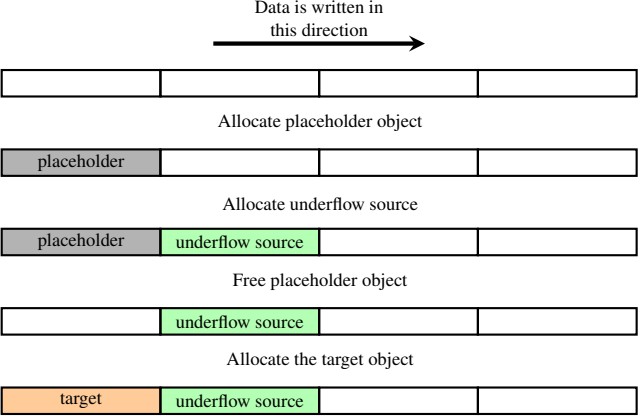

Figure 6: A solution for the reverse allocation order challenge without any noise.

of allocation sequences to choose from in all the challenges:

1. Allocate an object of the target caches size

2. Free a previously allocated object

3. Allocate the first target object (overflow/underflow source)

4. Allocate the second target object (overflow/underflow target)

For both of these kinds of challenges, we scale the difficulty by adding a certain number of noise allocations. For each "noise", when triggering the allocation of the overflow/underflow source additional objects are allocated before and after the overflow/underflow source. These noise allocations can not be freed by the algorithms, so it has to find a way to manipulate the heap surface in a way that circumvents them. We chose to always enable the algorithm to trigger allocations of a single object of the target cache size without any noise, as this is also almost always possible in real-world scenarios. For example, the `add_key` system call, which adds a new key to a specified key ring, will trigger an allocation of the size of the `payload` parameter on the kernel heap, so it can be used for defragmentation and manipulation of all sized caches without accompanying noise allocations [7, 28].

We ran the experiments on a virtual machine running Ubuntu 20.04, kernel version 5.9.7 with free list randomization disabled, a 16 core Intel Xeon Gold 6130 processor,

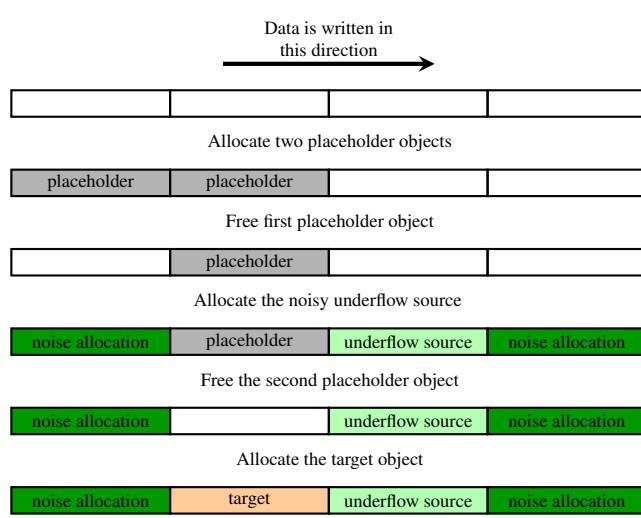

Figure 7: A solution for the reverse allocation order challenge with one noise.

and 64 gigabytes of RAM. We ran our experiments against the SLUB allocator, as it is the modern default. While being different internally the SLAB allocator would behave the same way in the context of our experiments, as its properties which are relevant for the manipulation process (sorting of objects into caches, usage and behavior of the free list etc.) are identical.

### 7.1.1 Pseudo-Random Search

To evaluate the pseudo-random search we ran it on the previously described challenges with increasing noise. For each challenge, we ran it 100 times, with an upper limit of 200000 candidates being generated. While a larger number of candidates may lead to more successful runs, with one run taking up to 10 minutes, this was the maximum possible number given our computational resources. The allocation-free ratio was set to 0.5. This value was determined experimentally, as it showed the best results across all noise levels. We created benchmarks for noise values from ranging 0 to 5, resulting in 12 experiments (two per noise value, natural allocation order and reverse allocation order). The results are listed in table 1. As we can see in the challenges without noise, the "natural" challenge poses fewer problems to the algorithms than the "reverse" challenge. While this is as expected, it is interesting to see that with the addition of noise, the "reverse" challenge actually becomes easier than the "natural" challenge. The reason for this becomes apparent if we take a look at an example solution for the "reverse" challenge with one noise, which is shown in Figure 7. First, we allocate two placeholder objects. Then, we free the first one, placing that slot at the beginning of the free list. Then we trigger the allocation of the underflow source. This results in the first noise allocation being placed in the previously freed slot, and the placeholder object in front

| Noise | Solved Natural | Solved Reversed | Avg. Tries Natural | σ Natural | Avg. Tries Reversed | σ Reversed |
|---|---|---|---|---|---|---|
| 0 | 100% | 100% | 3 | 1.99 | 9 | 7.73 |
| 1 | 100% | 100% | 195 | 192.6 | 60 | 54.88 |
| 2 | 100% | 100% | 2601 | 1957.6 | 375 | 366.86 |
| 3 | 100% | 100% | 29648 | 29087.47 | 2795 | 2615.93 |
| 4 | 32% | 100% | (73900) | (46873.98) | 28408 | 27671.43 |
| 5 | 1% | 47% | - | - | (90961) | (60256.65) |

Table 1: Results of the synthetic benchmarks of pseudo-random search. For each number of allocation noise, the percentage of successful solves is given for both the natural allocation order and the reverse allocation order. Additionally, we listed the average number of tries needed in case of success and the standard deviation of the number of tries. The statistics in brackets are those where not all tries succeeded, so they have to be treated with care as they only represent the successful runs.

of the underflow source. By freeing the placeholder object afterward, we can again put this slot at the beginning of the free list. By triggering the allocation of the target object now, we achieve the desired layout. This is the shortest possible solution for this specific challenge. Looking at the "natural" version of this challenge, it requires at least one more allocation of a placeholder object to successfully manipulate the free list for the desired heap layout. Therefore, the algorithm has to find a somewhat more specific solution for this problem. This feature of the natural challenge (requiring more allocations/frees for a minimal solution) remains when we add more noise, and this is also reflected in the experiment results. While pseudo-random search starts to fail at four noise in the "natural" challenge, it still solves the "reverse" version of it 100% of the time.

In general, pseudo-random search performs reasonably well. In our setting, it solved all problems with up to three noise within the 200000 tries 100% of the time. However, we can see that the average number of tries it needs to succeed as well as the standard deviation of tries needed grows exponentially. To give some perspective regarding the execution time, running the experiment with five noise, which was only solved once in 100 tries, took about 10.5 hours to finish.

### 7.1.2   KEvoHeap

To evaluate KEvoHeap we ran it on the same set of challenges as the pseudo-random search. To generate an initial population we used the same pseudo-random generation method that pseudo-random search uses to generate its candidates. We used an initial population size of 400, $\mu$, and $\lambda$ values of 200 each, a mutation probability of 0.9, and a crossover probability of 0.1. These values are identical to the ones used in the evaluation of EvoHeap [33] (besides the initial population size) and proved to be reasonable in our experiments. The maximum number of mutations was set to 5. When choosing the kind of mutation to be performed, `Mutate` was chosen with a probability of 0.7, while `Spray`, `Hole Spray`, and `Shorten` were all assigned a probability of 0.1

each. These values were determined by strategic experimentation and proved to perform best. After some experiments, we also decided to disable the "Allocate in a loop" directive (see Section D.1), as in our scenario it seemed to rather bloat up the candidates (especially in combination with the `Spray` mutation) and would not reasonably contribute to finding a better solution. As the algorithm has only one allocation directive to choose from, there is only one size group, sub group, and selector. The upper generation limit was set to 1000. The results are listed in table 2.

As you can see, KEvoHeap's performance is a big improvement over pseudo-random search. As one generation consists of 400 candidates that are evaluated, the performance for noise values of 0 and 1 can be seen as identical to pseudo-random search, both for the "natural" and "reverse" challenge. This is not surprising, as the initialization routine is identical to pseudo-random search, and pseudo-random search found solutions for these problems with fewer attempts than the initial population size. As the noise grows, the number of average generations needed to solve the problem grows with it. What is interesting to see is that in the natural challenge, the average generations and the standard deviation grow exponentially, however, the average tries and standard deviation of the reverse challenge seem to grow linearly, with a slightly bigger jump at 6 noise. This could indicate that it is also an exponential rise, but with a very low base. KEvoHeap proceeds to solve all challenges up to 6 noise with way fewer candidates generated than pseudo-random search. This is also reflected in the runtime: While pseudo-random search needed about 10.5 hours for 100 runs with five noise and natural allocation order (without solving the problem most of the time), it took KEvoHeap only about 2 hours for the same problem, with a 100% success rate.

## 8   Analysis and Discussion

As stated before, KEvoHeap proved to be a vast improvement over pseudo-random search. Figure 8 illustrates the difference in terms of candidates being generated by both algorithms. While both algorithms perform equally at low noise levels,

| Noise | Solved Natural | Solved Reversed | Avg. Generations Natural | σ Natural | Avg. Generations Reversed | σ Reversed |
|---|---|---|---|---|---|---|
| 0 | 100% | 100% | 1 | 0 | 1 | 0 |
| 1 | 100% | 100% | 1.09 | 0.35 | 1 | 0 |
| 2 | 100% | 100% | 3 | 2.09 | 1.24 | 0.77 |
| 3 | 100% | 100% | 6.87 | 4.3 | 3.05 | 1.83 |
| 4 | 100% | 100% | 11.62 | 7.24 | 4.75 | 2.32 |
| 5 | 100% | 100% | 22.85 | 13.97 | 6.37 | 3.29 |
| 6 | 100% | 100% | 43.25 | 29.78 | 8.2 | 5.8 |

Table 2: Results of the synthetic benchmarks of KEvoHeap. For each number of allocation noise, the percentage of successful solves is given for both the natural allocation order and the reverse allocation order. Additionally, we listed the average number of generations needed in case of success and the standard deviation of the numbers of generations.

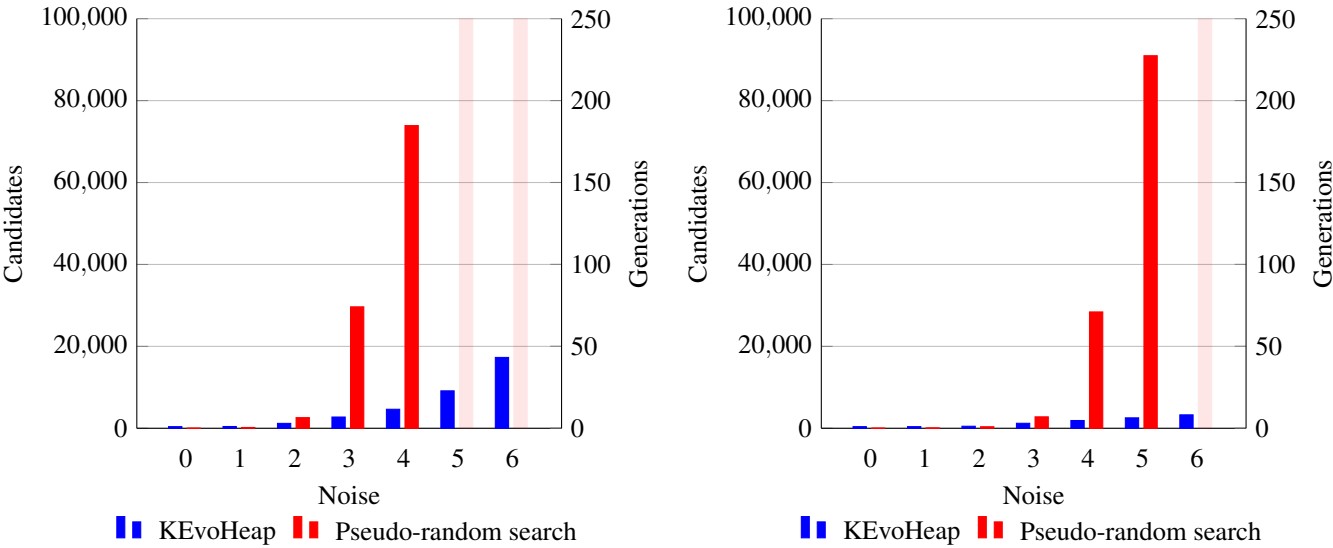

Figure 8: Bar chart showing the average tries/generations needed in both algorithms with respect to the level of noise. The left chart shows the results for the "natural" challenge, the right one shows results for the "reverse" challenge.

the number of candidates needed by pseudo-random search starts to skyrocket at three noise in the "natural" challenge and four noise in the "reverse" challenge. The number of generations also increases, but way slower. While in the "natural" challenge pseudo-random search only needs approximately twice as many candidates as KEvoHeap at two noise, it already needs almost 16 times as many candidates at four noise, and as only 37% of the experiments succeeded and we only look at the successful cases, the actual number is probably even higher. Another difference between both algorithms is the difference in the standard deviation. For pseudo-random search the standard deviation is always approximately equal to the mean, showing that the actual number of tries needed to solve a problem can vary a lot[2]. For KEvoHeap, the standard deviation is always about half the mean, which indicates

that it is more stable runtime-wise. This difference between the two algorithms is also illustrated by example for the "reverse" challenge with three noise in Figure 9. The box plots clearly show the difference in stability regarding the number of candidates that have to be generated. While the number of candidates has its median at 1933.5 for pseudo-random search, numbers go even up to 12042, and from the quartiles, you can see that the results are widely distributed. In contrast, the results from KEvoHeap are way closer together, and there are no extreme outliers. The superior results of KEvoHeap outline the advantages of a structured approach as opposed to a (pseudo-)random approach. The fact that candidate solutions can be improved in small steps by changing small parts of it makes the fitness function very smooth, which is a good property for a successful genetic algorithm [20]. While the algorithm still contains plenty of random components, their design and the selection process guide the algorithm towards better solutions. This results in the improved results, execution time, and stability.

---

[2]The results of the experiments where pseudo-random search failed at some tries do not reflect this property, as the failed cases were not taken into account when calculating the mean/standard deviation, so these results are not representative for the actual performance.

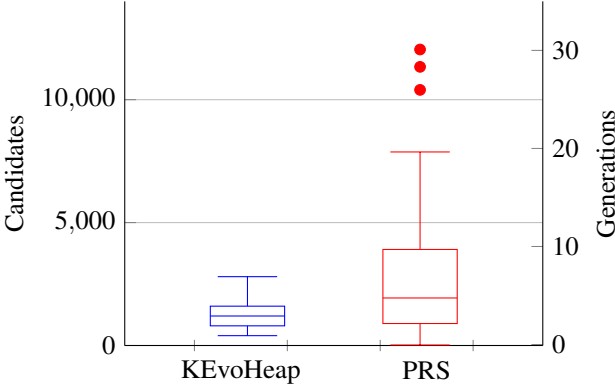

Figure 9: Box plot illustrating the distribution of numbers of candidates generated for the "reverse" challenge with three noise.

When comparing our results to the benchmarks created by Heelan for heap layout manipulation in user space [33–35] we can see that both algorithms perform way better when targeting the kernel heap. In our setting even pseudo-random search can solve all problems up to three noise with reasonable effort. In user space, pseudo-random search was able to solve the problems without noise most of the time, but performance dropped rapidly when any noise was introduced. In the kernel, we can see that the introduction of noise increases the number of candidates that have to be generated exponentially, but real problems only occur if multiple noise allocations are present. EvoHeap drastically improved the success in user space with an average 95.3% success rate across all synthetic benchmarks and all targeted allocators with up to four noise. In the kernel, KEvoHeap, our variant of EvoHeap, even surpassed the user space results with a 100% success rate with up to six noise. We did not run experiments with more than six noise allocations as we evaluated this as an unrealistic setting to appear in the real world, but the rate of increase in required generations suggests that more noise can theoretically still be added before we have to increase the generation limit. The success of KEvoHeap can be attributed to certain properties of slab allocation. The process of sorting objects of certain sizes in assigned memory regions (or caches) can make it harder to pair a vulnerable object with a suitable target, as both have to be allocated in the same cache. However, in terms of heap layout manipulation, it enables the previously described defragmentation, which is a big advantage from an attacker's point of view. With defragmentation, we can disregard all allocations that were performed previously to our attack by forcing the creation of a new empty slab. This is in many ways advantageous, as the problem is reduced to circumventing the accompanying noise allocations of the overflow source and target. When tackling problems in real-world scenarios in the future it will also be a big advantage that it is almost always possible to trigger single allocations in a targeted cache, e.g. with `add_key` as previously described (see Section 7.1). Due to this simplification, the question

arises if an optimization approach like a genetic algorithm might be unnecessarily sophisticated. Even for user space targets, Heelan mentioned that he considered simulated annealing as an alternative to genetic algorithms but disregarded it as genetic algorithms according to the literature proved to create better results at the cost of higher computational effort [33]. As the problem in kernel space seems to be easier to solve, it might be good to reconsider simulated annealing to further reduce the computational effort.

# 9  Application in Realistic Settings

As we saw in the evaluation on the synthetic benchmarks, KEvoHeap can be very effective at crafting desired heap layouts. However, there are still some challenges to overcome to make it applicable to real-world scenarios. In this section, we will present some of them and propose possible solutions for them. Finally, we present the exploitation of a vulnerable kernel module that serves as a case study for real-world scenarios.

## 9.1  Accessing Distance of Target Objects

In the Kernel-SIEVE framework, the kernel module gives us access to information about the current state of the kernel heap. We use this to calculate the distance of the two target objects, which serves as feedback to KEvoHeap or any other potential algorithm. In a realistic scenario, where our candidates consist of several subsequent system calls, we do not have this feedback, and without this, an optimization algorithm can not work. Additionally, a system call might allocate not one, but multiple different objects in one or multiple caches, so initially it is not even clear which object is the relevant one, be it overflow source or target. To tackle this, we have to specify which is our object of interest and find a way to extract the information needed for the feedback loop. After some experiments we concluded that this problem can be solved by using one of the several tracing mechanisms the Linux kernel provides. In particular kprobes [2], bpf-trace [48], and ftrace [3] seem to be viable tools for this, and we successfully implemented solutions using both kprobes and bpftrace respectively.

## 9.2  Instability of Results after Multiple Executions

In Section 5 we explained that additional problems arise when we try to execute multiple manipulation attempts in a row. One of the main problems is that subsequent attempts influence each other because they reorder the free list of the target slab. In Kernel-SIEVE we deal with this by creating a separate cache that we have full control over, so we can destroy and recreate it between each candidate execution. When we

deal with a real vulnerability we do not have this kind of control, as the target objects (and the accompanying noise) will be placed in the designated caches. The main problem that we have is that the kernel is in a different state when we executed one candidate than it was before, and this state influences the following candidates. So what if we could execute one candidate, and then reset the state of the kernel to the way it was previous to the execution? This would solve all our problems, as now each candidate gets executed with an identical starting state. In reality, something similar is possible. QEMU [12] is a widely adopted open-source emulator and virtualizer that among other things can be used to run an operating system as a virtual machine. It comes with a feature to create snapshots that not only save the current disk state, but also the RAM and CPU state, which can be used to restore the complete state of a system at a specific point in time [4]. Using this we could reset the VM after each candidate. Of course, this brings some performance drawbacks, but as KEvoHeap proved to be very efficient in the synthetic benchmarks it is fair to assume that performance should still be acceptable. We see this approach as very promising, as it does not require additional per-case analysis and works equally for each target. However, it requires some structural changes. As the entire virtual machine gets reset in the process of evaluating the candidates, the algorithm has to be run outside of the virtual machine. The framework would have to trigger the executions of the candidates from the outside and extract the results before reset. We implemented a proof-of-concept prototype using this approach that confirmed our assumptions. Figure 10 illustrates how we decoupled the algorithm from the virtual machine that runs the candidates.

A runner script first sets up the QEMU virtual machine. This includes starting the virtual machine, loading the kernel module, and starting a simple bind shell using netcat [8] that we use later to issue commands, transfer candidates and extract results. Then we enter the main loop of the genetic algorithm. We first query KEvoHeap to generate a batch of candidates (either the initial population, or, on subsequent calls, the next generation). Then we transfer the whole batch to the VM using scp [9]. Now that everything is in place for the execution, we use QEMU-Monitor [13], a tool for managing running QEMU instances, to save the state of the virtual machine. To finally run the candidates, we use our bind shell to run each candidate individually. We extract the result in a similar way, and then reload our saved state for the next execution. With this approach we were able to solve heap layout manipulation problems with the actual sized caches, not relying on the custom cache mode. This came with a performance drop: While the number of generations needed to solve a problem did not change, executing one generation of 400 candidates took about 3 minutes. For the challenge with five noise and natural allocation order this results in an expected runtime of 69 minutes to solve a single problem. Running a benchmark like we did in Section 7.1.2

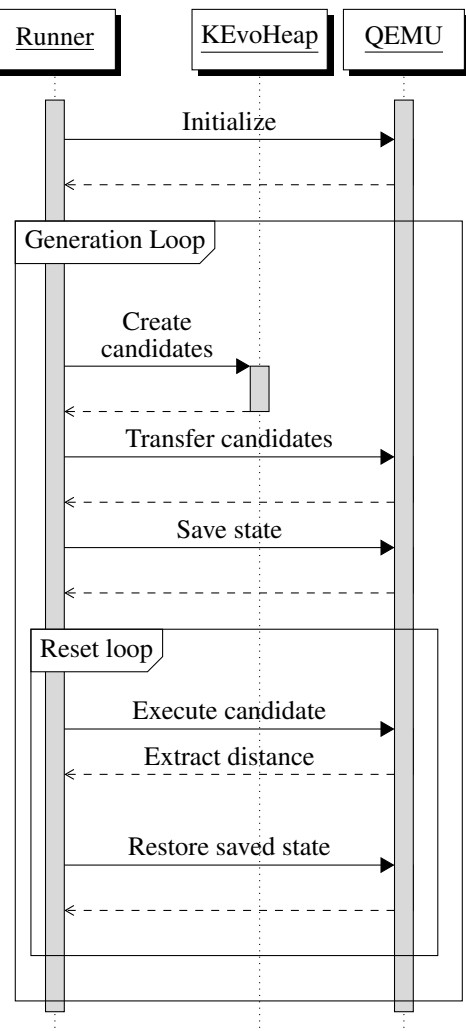

Figure 10: Sequence diagram for the prototypical solution utilizing QEMU's `savevm` feature.

with 100 runs would result in an estimated 115 hours. This is an enormous increase in runtime in contrast to the 2 hours it took with the custom cache mode, but one hour to find a valid solution is still within reason. Also it might be possible to further boost the performance, but this is beyond the scope of this work.

Figure 15 (cf. appendix D.5) provides a comparison of the runtimes of running 100 experiments using KevoHeap, pseudo-random search, and fast-reset-KEvoHeap. Notice that we used a logarithmically scaled y-axis, as fast-reset-KEvoHeap's runtime is significantly longer due to the significant performance drop that comes with the fast reset method. This makes it hard to visually compare to the other runtimes without scaling. Because of the very long execution times, we only estimated the runtimes of fast-reset-KEvoHeap using the average generations needed and the measured runtime per generation. From the plot, you can see that while the performance drop is large, because of the superiority of KEvoHeap

the runtime of pseudo-random search approaches the runtime of fast-reset-KEvoHeap rapidly with rising noise levels. The measurements of pseudo-random search lose meaning starting at four noise, as not all runs are successful, so the runtime to solve the 100 problems is actually higher. Keeping this in mind, the runtime of pseudo-random search probably supersedes the runtime of the fast-reset-KEvoHeap at five or six noise.

## 9.3 Application of KEvoHeap in real-world vulnerabilities

In order to prove the usage of our approach we created a vulnerable kernel module which contains a heap buffer overflow vulnerability. This kernel module serves as a case study for a real vulnerability in the Linux kernel.

Its behaviour is quite simple: It can be queried to allocate a buffer and to write/print data to/from said buffer. Before and after the buffer gets allocated noise allocations will be made to make exploitation non-trivial.

Our exploit performs heap layout manipulation to shape the heap into an exploitable state utilizing KEvoHeap. Internally KEvoHeap uses kmalloc and kfree to create the desired kernel heap layout. In order to find a solution for this, we or respectively an attacker need an identical vulnerable Linux kernel version with root privileges. This lab environment is running KEvoHeap to identify appropriate allocations and deallocations to bring the Linux kernel heap into an exploitable state. This means that the target objects are placed next to each other. To identify these allocations, bpftrace is used, which requires root privileges. Figure 11a

illustrates this in a simplified manner. The fact that the target kernel as well as the KEvoHeap kernel module and bpftrace are executed within QEMU is omitted on behalf of simplicity.

An attacker or respectively a security researcher can use these allocations and deallocations via various system calls. We use the shmget/shmctl system calls to allocate objects in the kmalloc-256 cache. These calls trigger (de)allocations of objects, which are of appropriate size. A suiting series of allocations and deallocations are then found with KEvoHeap. KEvoHeap will generate candidates consisting of calls to the kernel module and shmget/shmctl, which will then be inserted into a general corpus program which takes care of setup and teardown. The generated candidates are then build and executed.

Once KEvoHeap has found a solution, it can be used as part of the exploitation process to manipulate the heap into an exploitable state. Figure 11b outlines this.

By doing this, we have shown that KEvoHeap is able to automatically convert the kernel heap to a suitable state even in the case of real-world vulnerabilities. This exploit and the vulnerable kernel module can be found in our GitHub repository [43].

## 10 Conclusion

In this paper, we presented Kernel-SIEVE, a framework for evaluating heap layout manipulation algorithms that target the SLAB/SLUB allocator in the Linux kernel. With this, we created the (to our knowledge) first performance benchmarks for heap layout manipulation algorithms in the Linux

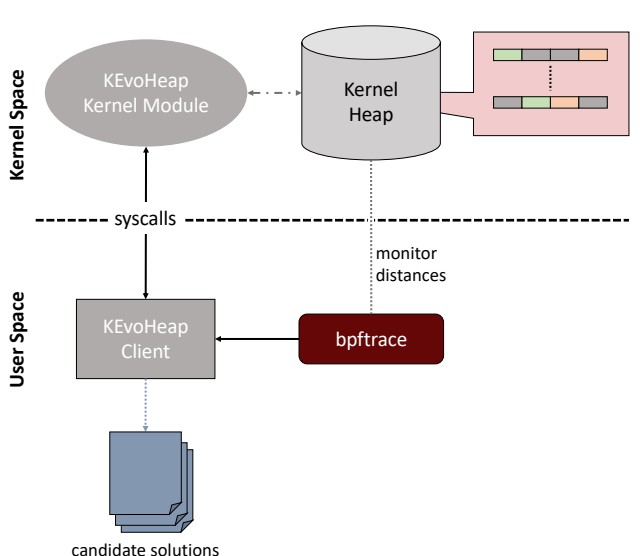

(a) Generate solutions for heap layout manipulations

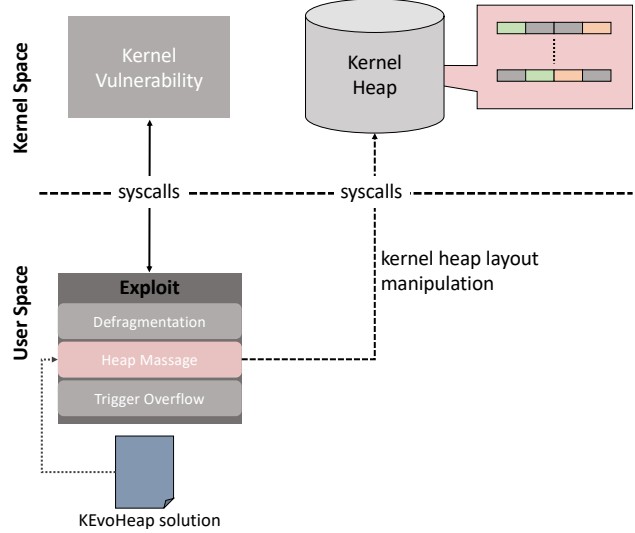

(b) Applying results from KEvoHeap in an exploit

Figure 11: Application KEvoHeap in real-world vulnerabilities

kernel. Moreover, we proposed and evaluated two algorithms: A pseudo-random search, whose performance serves as a baseline, and KEvoHeap, a genetic algorithm based on Heelan's EvoHeap [33, 35]. We have shown that KEvoHeap is successful at creating the desired heap layout in all test cases and also surpasses the user-space performance benchmarks of EvoHeap. Besides that, we discussed the challenges of applying these kinds of algorithms in real-world scenarios and weigh different possible approaches to tackle these problems. Finally we have shown the application of KEvoHeap in a real-world scenario utilizing our case study.

While further research into this topic is necessary we believe that this research has taken the art of automating kernel exploits one step further.

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

# Appendix

In the following, we first briefly describe the implementation of the SLUB allocator. Then the algorithms of Heelan's work are summarized. After that, we give detailed information about the different algorithm implementations used in this paper.

## A   The SLUB Allocator

We will quickly go over the implementation of the SLUB allocator, which replaced the original SLAB allocator as the modern default in Linux.

Christoph Lameter developed the SLUB in 2007 out of frustration over the SLAB implementation [37, 38]. His two main complaints were:

1. The many different queues used in the SLAB implementation made the code quite complex.

2. In larger systems, the number of queues and objects in these queues can grow exponentially, which wastes memory which is only tied up in control structures.

The goals of SLUB were to provide better scalability and performance, while at the same time simplifying the general slab structure. Additionally, SLUB also vastly extended the debugging capabilities of the slab allocator. Today, SLUB has replaced SLAB as the default allocator in the Linux kernel.

SLUB indeed simplified the general slab structure. In SLUB, a slab only consists of allocated and/or free memory chunks, without any metadata at the beginning. This reduces the need for padding for alignment.

The free chunks form a linked list which is used to keep track of them. On allocation, the first element in this list is simply returned and removed from the list. The kernel stores a pointer to the free list together with information about the number of blocks in use and the total number of objects in the page frame in the `struct page`. This is a control structure that the kernel assigns to each physical memory page. It provides general information about the context in which the page is used. SLUB also keeps an array of slabs that are associated with single CPUs to avoid cache line bouncing. If these are not being used, they are simply put back into the `partial` list. In contrast to SLAB, there is only one list that manages slabs per cache, the `partial` list. Full slabs are simply forgotten until they become partial again. This reduces the aforementioned memory complexity which could become problematic in SLAB. Figure 12 illustrates these structures [25, 37, 38].

## B   Heelans work

The main work we build on is Sean Heelan's Ph.D. thesis about "Greybox Automatic Exploit Generation for Heap Overflows in Language Interpreters" [33]. Heelan breaks the problem of automatic exploit generation into several sub-problems, one of them being the Heap Layout Problem, which he also already addressed in previous publications [34, 35]. For this, he developed a framework for evaluating heap layout manipulation algorithms on different allocators called SIEVE. SIEVE consists of a driver that can be linked in combination with any allocator that exposes the standard allocation interface from the ANSI C standard [16], which are most importantly the functions `malloc`, `free`, `calloc`, and `realloc`. The driver takes a file as input that consists of a set of directives that instruct the driver to perform certain allocation operations. The directives are of the following form:

1. `<malloc size ID>`

2. `<calloc nmemb size ID>`

3. `<free ID>`

4. `<realloc oldID size ID>`

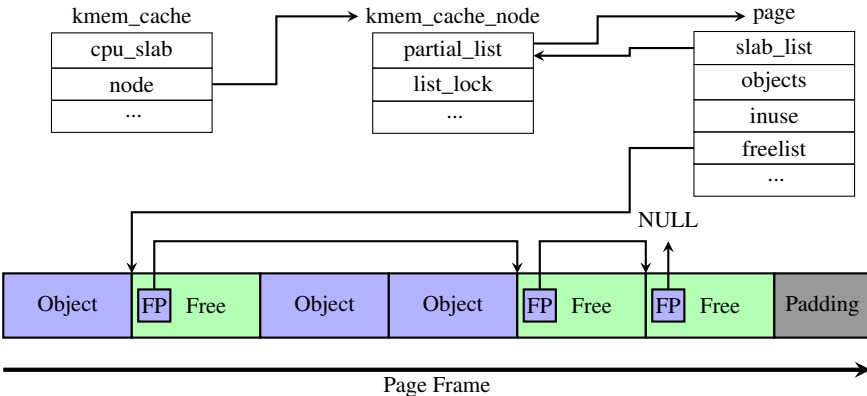

Figure 12: Structure of a slab used by the SLUB allocator.

5. `<fst size>`

6. `<snd size>`

The first four directives are used to trigger the standard memory allocation functions. The ID parameter assigns an ID to the allocations, which can be used to reference them again in a `free` or a `realloc`. The last two directives trigger the allocation of an object of a given size, which can be used to simulate an overflow/underflow source and a target. After executing the given directives, the driver returns the distance of the target allocations. With this, you can create challenges with the objective to place the target objects relative to each other at a certain distance. Allocation noise can be simulated by creating interactions from multiple directives that cause side effects. For example, instead of letting an algorithm simply allocate the overflow source, we can add an allocation of an object right after the source, which the algorithm then has to circumvent for being successful. The SIEVE framework also offers a Python API that helps with the creation of different kinds of challenges.

Heelan provided two algorithms for solving the heap layout manipulation problem: A pseudo-random search, and a genetic algorithm called EvoHeap. Algorithm 3 shows how the pseudo-random search generates candidates and executes them.

In each iteration of the loop in the `Search` function, the algorithm assembles a new candidate and executes it. If the distance returned equals the target distance, the candidate is returned. If after a set number of iterations no solution can be found, it returns *None*. The `ConstructCandidate` function takes care of the pseudo-random generation of candidates. It takes two arguments: The maximum length of a candidate *m* and a parameter *r* that probabilistically weights the number of allocations and frees performed in the candidate. First, the length of the candidate is determined by taking a random number between 1 and *m* (line 11) and a random index for the allocation of the first target object (line 12). Then it iterates the indices of the candidate. At the previously picked *fstIdx* the algorithm inserts the provided sequence to allocate the overflow source (line 15). On any other index, the algorithm either appends an allocation sequence or a free sequence, depending on a probability-based on *r*. The `AppendAllocSequence` and `AppendFreeSequence` pick one of the available allocation/free sequences with equal probability. The `AppendFreeSequence` function redirects to `AppendAllocSequence` if no allocation is available for freeing. At last, the algorithm appends the sequence for allocating the target object (line 21) and returns the candidate.

While this algorithm performs almost no search strategy, it still produces some good results. Heelan evaluated it on three different allocators (avrlibc [1], dlmalloc [40] and tcmalloc [30]) with synthetic challenges. The core of the challenges was to place the two objects directly adjacent, either normal or reversed. Additional experiments were performed with

---

```
1  Function Search(g, d, m, r):
2      for i ← 0 to g − 1 do
3          cand ← Init(ConstructCandidate(m,
              r))
4          dist ← Execute(cand)
5          if dist = d then
6              return cand
7      end
8      return None

9  Function ConstructCandidate(m, r):
10     candidates ←
           Init(GetStartingState())
11     len ← Random(1, m)
12     fstIdx ← Random(0, len − 1)
13     for i ← 0 to len − 1 do
14         if i = fstIdx then
15             AppendFstSequence(cand)
16         else if Random(0, 100) ≤ r then
17             AppendAllocSequence(cand)
18         else
19             AppendFreeSequence(cand)
20     end
21     AppendSndSequence(cand)
22     return cand
```

**Algorithm 3:** Pseudo-random search algorithm to find a solution that places two target objects at a certain distance in memory. *g* is the number of candidates to try, *d* the target distance, *m* the maximum size of the candidates, and *r* the allocation-free ratio [33].

the addition of allocation noise in the form of allocations before and after the allocation of the first target object, i.e. the overflow source. The starting state of the candidates was taken from the initialization routines of Python, PHP, and Ruby. The pseudo-random search was actually able to solve the majority of challenges as long as no noise was added with $g = 500.000$, $r = 98$, and $m = 1000$ . For avrlibc, it even solved all challenges without noise (normal and reversed), and 99% of the challenges using dlmalloc. The success rate dropped drastically when noise was introduced. When adding 4 noise allocations before and after the overflow source, the success rate averaged at 37% and went down to 17% at dlmalloc for the reversed challenge.

As an alternative to a simple pseudo-random search, Heelan proposed EvoHeap, a genetic algorithm for the heap layout manipulation problem.

### B.1    EvoHeap

The main routine of EvoHeap is shown in Algorithm 4. First, an initial population is generated randomly (line 2) and gets

evaluated (line 3). If this initial population already contains a valid solution, the population and their fitness values are returned (line 5). If not, the main cycle is entered. In each generation, λ children are generated from the existing population via the GetChildren function and get evaluated (lines 7+8). The GetChildren function takes the mutation probability *mxpb* and the crossover probability *cxpb* and creates the offspring by randomly selecting parents from the population and performing mutations and crossovers based on the provided probabilities (see Algorithm 5). The internals of the mutation operator will be explained in detail in chapter 6.2 when we present our modified version of this algorithm for the kernel. For now, it is enough to know that it can either insert or remove allocation/free sequences from a candidate in different ways. The crossover operator implements some kind of two-point crossover which will also be gone into some more detail in chapter 6.2. If one of the children contains a valid solution, the children and their fitness values are returned (line 10). Otherwise, a new population for the next generation gets selected via a $(\mu + \lambda)$ selection strategy using a mixture of elitist and double tournament selection. Details on this will also be given in chapter 6.2, as we use the same selection strategy as EvoHeap. Then the cycle starts from the beginning and a new generation is entered. If after *g* generations no solution has been found, the current population is returned with their corresponding fitness values.

```
1  Function EvoHeap(g, popsz, μ, λ, mxpb,
     cxpb):
2  |   pop ← InitialisePopulation(popsz)
3  |   popF ← Evaluate(pop)
4  |   if SolutionFound(popF) then
5  |   |   return pop, popF
6  |   while g > 0 do
7  |   |   ch ← GetChildren(pop, λ, mxpb,
     |   |     cxpb)
8  |   |   chF ← Evaluate(ch)
9  |   |   if SolutionFound(chF) then
10 |   |   |   return ch, chF
11 |   |   pop, popF ←
     |   |     Select(μ, pop + ch, popF + chF)
12 |   |   g ← g - 1
13 |   end
14 |   return pop, popF
```

**Algorithm 4:** The main routine of EvoHeap. *g* is the maximum number of generations to run. *mxpb* and *cxpb* are the mutation/crossover probabilities.

EvoHeap showed significant improvements over the pseudo-random search during evaluation. On average, Evo-Heap solved 95.3% of all of the synthetic challenges, while

```
1  Function GetChildren(pop, λ, mxpb,
     cxpb):
2  |   children ← []
3  |   while λ > 0 do
4  |   |   parentA ←
     |   |     pop[Random(0, len(pop))]
5  |   |   r ← Random(0, 1)
6  |   |   if r < mxpb then
7  |   |   |   new ← Mutate(parentA)
8  |   |   else if r < mxpb + cxpb then
9  |   |   |   parentB ←
     |   |   |     pop[Random(0, len(pop))]
10 |   |   |   new ← Crossover(parentA,
     |   |   |     parentB)
11 |   |   else
12 |   |   |   children.append(parentA)
13 |   |   λ ← λ - 1
14 |   end
```

**Algorithm 5:** The routine that creates λ offspring from the population *pop* using the mutation probability *mxpb* and the crossover probability *cxpb*.

pseudo-random search solved about 51% of all challenges. Heelan labeled certain specific challenges as "very-hard", of which pseudo-random search was only able to solve about 8% of. EvoHeap was able to solve 80% of these challenges, again showing the massive improvement it brought. Both algorithms were also evaluated in a realistic scenario, where they should solve heap layout manipulation problems in the PHP interpreter. EvoHeap here also surpassed pseudo-random search by solving 84.2% of the problems on average, in contrast to 61% solved by random search. In addition to the quality of the solutions, EvoHeap was also faster at finding these solutions most of the time. In the synthetic benchmarks, EvoHeap was faster 74% of the time, and pseudo-random search was only faster on problems that Heelan labeled as "very-easy". On the PHP benchmarks, EvoHeap was always faster than pseudo-random search (only considering problems that both algorithms solved), with a time difference averaging at 600 seconds.

## C  Pseudo-Random Search

In the following we show our implementation of the pseudo-random search with minor adjustments to the one used in Heelan's work (cf. Algorithm 6).

The search is made pseudo-random by the *r* parameter that defines the ratio of allocations to frees. In our experiments, 90 proved to be a good value if noise is low, but it depends on the challenge. With increasing noise, a lower value can be useful to increase the number of

**Input:** $g, m, r$
**Output:** A set of candidate solutions

```
 1 Function GenerateBatch(g, m, r):
 2     candidates ← []
 3     for i ← 1 to g do
 4         candidates.append(
 5             ConstructCandidate(m, r)
 6         )
 7     end
 8     return candidates

 9 Function ConstructCandidate(m, r):
10     len ← Random(1, m)
11     fstIdx ← Random(0, len − 1)
12     for i ← 0 to len − 1 do
13         if i = fstIdx then
14             AppendFstSequence(cand)
15         else if Random(0, 100) ≤ r then
16             AppendAllocSequence(cand)
17         else
18             AppendFreeSequence(cand)
19     end
20     AppendSndSequence(cand)
21     return cand
```

**Algorithm 6:** Method to pseudo-randomly generate a batch of candidate solutions. $g$ is the total number of candidates to generate. $m$ is the maximum number of directives per candidate. $r$ is the ratio of allocations to frees.

frees generated. To customize the challenges we have to implement the `AppendFstSequence`, `AppendSndSequence`, `AppendAllocSequence`, and `AppendFreeSequence` functions. As you can see in comparison to Heelan's implementation of pseudo-random search [33], we left out the `GetStartingState` function, which can be used to create a starting heap configuration, emulating previous allocations. In the case of heap layout manipulation in the kernel, this would not increase the difficulty of the challenge, as we could also always just perform defragmentation again to flatten the heap surface.

The `AppendFstSeqence`/`AppendSndSequence` functions should return the respective sequences for allocating the first/second target object. The simplest form would be the simple instruction `"fst <size>"`/`"snd <size>"`, but they can also consist of multiple other directives to simulate noise. The `AppendAllocSequence` function should choose between available allocation sequences with equal probability. The `AppendFreeSequence` function should choose a random sequence from the available free sequences. If there is no allocation available to free, it simply redirects the call to `AppendAllocSequence`.

## D  KEvoHeap

In this section, we will explain how KEvoHeap represents its individuals and how each of the genetic operators was implemented.

### D.1  Individual Representation

For KEvoHeap we use the same individual representation that Heelan used in EvoHeap. The individuals are designed in a way that decouples them from the actual code of the candidate solution.

Each interaction that is available to the algorithm gets mapped to a representative ID[3]. The algorithm then only acts on these IDs. That means that the user has to provide a mapping function that translates these IDs to their counterpart instructions and vice versa. By this, the algorithm does not need to know specifics about the problem it should solve. Figure 13 illustrates the translation cycle performed by the algorithm.

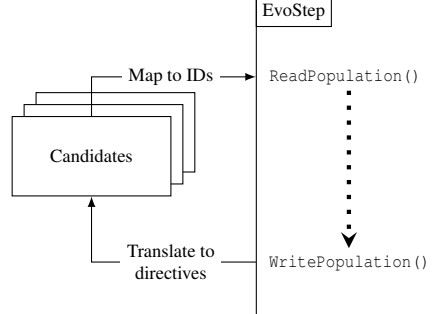

Figure 13: The translation cycle of the genetic algorithm.

At the beginning of each evolution step, the algorithm uses the user-provided mapping to translate the directives of the candidates to their respective IDs. When the new population is written at the end, they are translated into actual directives in a similar fashion.

Each individual represents a candidate solution that is made of multiple directives. The algorithms represent the directives as 128-bit integers. Figure 14 shows how the coding works. The first 8 bits define the type of the directive. The interpretation of the rest depends on the type.

- **Allocate:** The `Allocate` directive represents an interaction sequence that results in the allocation of an object. It is very likely that we do not have one single primitive to trigger allocations in the target cache but can choose between multiple different, which are also different in behavior and quality. For example, we might have one

---

[3]As you will see later, it is actually not just a simple ID but they are also sorted into groups to provide some hierarchies between different primitives, but to illustrate the general principle we can just assume that each interaction gets mapped to a simple ID.

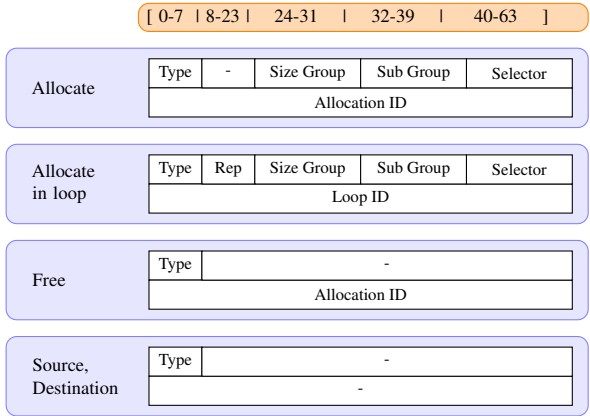

Figure 14: The representation of the directives after Heelan [33]. Each directive is represented by a 128-bit integer. The type field is always the first, the following fields depend on the type.

primitive that triggers a single allocation in the target cache, and a second one that triggers multiple allocations at once. Most of the time we would probably prefer to use the primitive that allows us more granular control about the allocations made. But in certain situations a primitive that triggers multiple allocations can be very useful, e.g. to "flatten" the heap structure after a noisy previous allocation. To tackle these different kinds of allocation primitives they can be sorted into groups and sub-groups, which themselves can be assigned probabilities. When e.g. a new allocation is generated through mutation (as we will see later), the kind of allocation will be drawn according to the given probability distribution. The first group, the group of the highest order, is called the "Size Group". In Heelan's algorithm that targeted user space allocators, this group was used to differentiate between primitives that trigger allocations of objects of different sizes. Due to the nature of slab allocation, this kind of grouping is not applicable here, as an allocation of diverging size would either be placed in a different cache or if the difference is small enough to still be placed in the same cache, would still behave the same way as all other allocations, as all allocations in a cache are placed in a chunk of equal size (as described in Section 4.1)[4]. We still decided to keep the group structure the same, as it allows very fine control between different primitives. If the user decides that this fine control is not necessary, he can simply provide only one group. Here, the "Size Group" can differentiate between allocations of one or multiple objects. Each "Size Group" is divided into one or multiple "Sub Groups".

---

[4]As mentioned before, Linux also offers the SLOB allocator as an alternative. Here objects are not stored in caches but a simple free list. While the SLOB allocator is out of scope for this work, a "Size Group" in the original sense would be applicable here, as objects of different sizes can be placed next to each other.

These are just another layer of control for defining a probability distribution over interaction sequences of the same size but different quality. The actual primitive used is finally selected via the "Selector" field. Each allocation is then assigned a 64-bit "Allocation ID". This ID can be referenced by a "Free" directive to free said allocation. By using 64-bit for generating the ID we can simply generate these IDs randomly while mutating or recombining individuals without having to worry about ID collisions.

- **Allocate in a loop:** The `Allocate in a loop` directive simply performs an "Allocate" directive multiple times. The fields are identical to those of the "Allocate" directive with an additional "rep" field that defines how often the allocation should be repeated.

- **Free:** The `Free` directive triggers a free of a previous allocation. The allocation is referenced by the 64-bit allocation ID.

- **Source/Destination:** The `Source`/`Destination` directives trigger an allocation of the Source/Destination object. They require no additional parameters.

## D.2 Mutation

As described before, the mutation is one of the basic operations a genetic algorithm performs to generate offspring by altering a single individual. Our mutation operator is very similar to the one used in EvoHeap [33]. When the `Mutate` function is called from within `GetChildren` (Algorithm 18), a number of mutations to be applied is drawn from a geometrically decreasing probability distribution between 1 and a maximum set by the user.

For each mutation to be applied the algorithm chooses randomly[5] from one of the following available operators:

- **Mutate:** The `Mutate` operator first selects a random number of `Allocate` and/or `Free` directives. Each of the selected `Allocate` directives are with equal probability then either changed to an allocation using a different primitive or to a `Free` of a previous allocation (if possible). A `Free` directive on the other hand is either changed into an `Allocation` directive using a random primitive drawn from the provided probability distribution of the groups and selectors or is changed to free a different previous allocation (if possible).

- **Spray:** The `Spray` operator inserts a new sequence of `Allocation` directives into the individual at a random offset. The primitives used in the allocation are drawn from the provided distribution but are all identical. The length of the sequence is randomly drawn from an interval provided by the user.

---

[5]Based on a probability distribution provided by the user.

- **Hole Spray:** The `Hole Spray` operation first generates a sequence of identical allocations just like the `Spray` operator but follows it up with a series of `Free` directives which free every second allocation made in the sequence. The length of the sequence is also drawn randomly from a user-provided interval, just like in the `Spray` operator. The combined sequence is placed at a random position in the individual.

- **Shorten:** The `Shorten` operator simply removes a contiguous section of directives from the individual.

EvoHeap also uses two additional operators: `Allocation Nudge` and `Free Nudge`. These two are alternate versions of the `Spray` and `Hole Spray` operators which only differ in the maximum length of the generated sequences. The Nudge-Operators should generate short sequences, while the normal ones could also generate sequences of very large length. We chose to remove this kind of differentiation in KEvoHeap for the following reason: As we apply defragmentation to the kernel heap right before we execute the candidate, we start allocating objects in an empty slab. That means, that most likely all our allocations will take place in one single page frame. If we allow the algorithm to make a huge number of allocations, we will most certainly exhaust the slab, which will cause the creation of a new slab on a new page frame. This will probably result in the second target object being allocated on a different page frame than the first one, so the target objects will not be adjacent to each other[6]. Because of this, we only allow sequences to be generated that will not exhaust the complete slab. As this leaves us with a rather small maximum length, we do not need a second operator for a different length class.

## D.3 Crossover

The crossover is the other operation a genetic algorithm uses to create offspring. Just like in EvoHeap we use a modified version of a two-point crossover [33, 57]. In the classic two-point crossover, two individuals of equal length swap a sequence between two set offsets. To cope with the fact that our individuals can be of different lengths, we select a sequence of random length starting from a random offset in each individual and swap them. This can possibly create invalid individuals, e.g. by removing an allocation from an individual that is freed later on or removing the allocation of the source or the destination. While it is possible to simply let the algorithm filter these in the selection process, it is more efficient to prevent this from happening. For this, we employ

---

[6]Technically this is not always true, as overflowing over page borders is possible and can lead to successful exploitation [46]. As SLUB does not keep metadata at the beginning of a slab, the objects could theoretically even be directly adjacent. However, this would require manipulating the buddy allocator in addition to the slab allocator. While this is possible, it is much more reasonable to assume that we can find a simpler solution inside the slab.

some kind of "housekeeping" routine. If we by accident remove the allocation of one of the target objects, we add it again manually. Additionally, we check the individual for double frees or frees of IDs that have not been allocated, and remove those.

## D.4 Selection

At the beginning of each `EvoStep` that is not the first the existing population has to be evaluated. The `Evaluate` function takes the current population and the distances that were returned from the client and assigns each individual a fitness value that we try to *minimize*. The fitness is calculated according to the following formula:

$$fitness(d) = \begin{cases} 2^{64} & \text{if } d \text{ is "error"} \\ 2^{64-1} & \text{if } d > 0 \\ abs(d) & \text{otherwise} \end{cases} \quad (1)$$

The distance that the client returns is calculated as ($srcAddr - dstAddr$), where $srcAddr$ is the address of the first target object, and $dstAddr$ the address of the second object. If the client returns an error the fitness of the individual is set to the maximum, $2^{64}$. If the distance is larger than zero the objects have been allocated in the wrong order. We set the fitness to $2^{64} - 1$ so we can distinguish them from execution errors. Otherwise, we simply set the fitness to the absolute value of d. This calculation assumes that we try to allocate the target *after* the source, simulating an overflow. If we want to simulate an underflow, the target has to be allocated *before* the source. This can be achieved by simply changing the second condition from $d > 0$ to $d < 0$.

The `Select` function is listed in Algorithm 7. It is identical to the one Heelan used in EvoHeap [33]. First, we divide the population into two groups: The *noerr* group contains all individuals that did not result in an error. The *orderok* group contains all groups that did not cause an error and also allocated the target objects in the correct order, so have a fitness lower than $2^{64} - 1$. If the *orderok* group is not empty, the offspring will be selected from it, otherwise, we select from the *noerr* group. In the unlikely case that all executions resulted in an error, we select from the whole population. For selection we use a mix of *elitist selection* and *double tournament selection*, which are both standard genetic algorithm selection functions [42, 57]. The weighting between both strategies is set via the *e* parameter. The third parameter of `SelBest` and `SelDoubleTourn` tells how many individuals should be selected. Elitist selection simply returns the $\mu \cdot e$ individuals with the best fitness. Double tournament selection is an a bit more sophisticated approach for selection. In standard tournament selection, a random set of individuals is taken from the population, from which the best according to their fitness is selected. This process gets repeated as often as the number of individuals that should be selected. In double tournament selections, the individuals have to pass *two*

tournaments: First, two individuals are selected via standard tournament selection. These two then participate in a *parsimony* tournament, in which the shorter individual is selected according to a user-provided probability between 0.5 and 1. This selection strategy puts a penalty on very long individuals. As through mutation individuals can grow rapidly in size, this strategy proved to counterbalance this growth [33, 42]. In our experiments we saw that it is most effective to use a value of 0.5 for $e$, splitting the selection in half between elitist and double tournament selection. The code for both algorithms is available on GitHub [43].

---

**1** **Function** Select(*pop, fit, μ*):
**2**     $noerr \leftarrow [], noerrFit \leftarrow []$
**3**     $orderok \leftarrow [], orderokFit \leftarrow []$
**4**     $i \leftarrow 0$
**5**     **while** $i < $ len(*pop*) **do**
**6**         **if** $fit[i] \neq 2^{64}$ **then**
**7**             $noerr$.append($pop[i]$)
**8**             $noerrFit$.append($fit[i]$)
**9**             **if** $fit[i] \neq 2^{64} - 1$ **then**
**10**                 $orderok$.append($pop[i]$)
**11**                 $orderokFit$.append($fit[i]$)
**12**         $i \leftarrow i + 1$
**13**     **end**
**14**     **if** len(*orderok*) $> 0$ **then**
**15**         $pop \leftarrow orderok$
**16**         $fit \leftarrow orderokFit$
**17**     **else if** len(*noerr*) $> 0$ **then**
**18**         $pop \leftarrow noerr$
**19**         $fit \leftarrow noerrFit$
**20**     $e \leftarrow$ GetFracElitism()
**21**     $b \leftarrow$ SelBest(*pop, fit, μ·e*)
**22**     $r \leftarrow$ SelDoubleTourn(*pop, fit,* $\mu \cdot (1-e))$
**23**     **return** $b + r$

**Algorithm 7:** The selection routine [33]. *pop* is the population, *fit* their fitnesses, and *μ* the number of individuals to select. It implements a $(\mu + \lambda)$ strategy, as both the parents and the children are taken into consideration.

## D.5  Evaluation Fast-reset-KEvoHeap

Figure 15 provides a comparison of the runtimes of running 100 experiments using KevoHeap, pseudo-random search, and fast-reset-KEvoHeap.

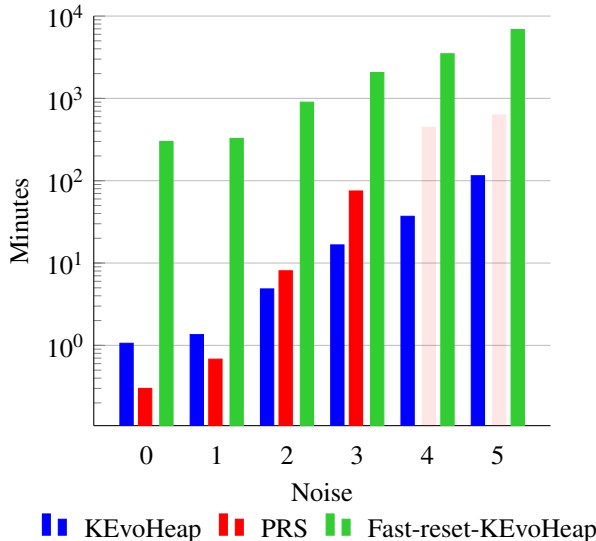

Figure 15: Bar plot illustrating the execution time of running 100 experiments with natural allocation order and different noise levels across KEvoHeap and pseudo-random search with the custom cache implementation, and KEvoHeap utilizing the fast reset strategy. The values for fast-reset-KEvoHeap were estimated using the average generations needed for KEvoHeap to solve the respective problems.

