# OpenReview forum: "Algorithmic Heap Layout Manipulation in the Linux Kernel"
_JSYS/2022/May_Papers — Submitted to JSYS May 22_

### Official Review · Reviewer_FC57 · 2022-05-31
**Review of "Algorithmic Heap Layout Manipulation in the Linux Kernel"**

**Decision:**

Weak reject: interesting papers with flaws, not sure if they can be fixed in three months

**Review:**

Paper Summary
=============

The authors analyse the problem of heap layout manipulation for the linux kernel and attempt to find solutions to automate this step. The paper starts with an analysis of the problem in general and of the specific challenges faced when solving this problem for the linux kernel. The authors go on to review the design of kernel memory management and then present their framework as an extension of the work from Heelan et al. The paper presents two algorithms (also inspired from prior work) and benchmarks their effectiveness against the allocator in the linux kernel. The authors present some concluding remarks and discussions.

Strengths
=========

 - the paper is well written
 - the problem is relevant and worthy of attention and research
 - the paper is very informative: it contains a useful introduction to the problem and to the memory management internals of the linux kernel

Weaknesses
==========

Thanks for your submission. I'm happy to see activity in this space: the problem you are investigating is interesting, relevant and timely. The way you approach the problem is also the right one in my opinion, and I would like to see this paper published.

In order to get there, in my opinion, you should focus on the following areas:

 - Tighter coupling to the linux kernel: I would recommend to take one extra step and reason in terms of system calls rather than in terms of invocations of kmalloc or analogous dynamic memory management API in the kernel. Basically, with Heelan's work in mind [33], I think the value of the paper would be immensely higher if you worked towards a SHRIKE (section 3.2 of [33]) for the kernel. My request could be motivated and justified by the following observations: 1) without this aspect, one is left to wonder whether you have "only" built (*) a new allocator to use SIEVE with (on top of avrlibc, dlmalloc and tcmalloc). 2) one is also left to wonder whether you could have achieved the same benchmarking setup by extracting the relevant dynamic memory management code of the kernel, built it in userspace and just used the original SIEVE framework (https://github.com/SeanHeelan/HeapLayout). I would recommend to invest some work into what you propose in section 9.1: analyse which system calls trigger which allocations and drive KSIEVE through a set of system calls rather than just invocations of kmalloc.

 - complete the kernel memory management picture: there are some tradeoffs when using kmalloc directly vs. using kmem_cache_alloc and similar calls. Also, there are generic slabs vs dedicated caches, vmalloc, etc. Given the excellent intro you have provided to kernel memory management, I would encourage you to complete the picture which is surely going to be an interesting reference for readers interested in learning these aspects.

 - exploit more deeply the fact that you're using the kernel allocator. As you point out at the end of section 4.2, the SLUB allocator has several peculiarities that make it unique and very different from dlmalloc, tcmalloc etc. I would have liked to see these considerations feature more prominently in your analysis and in the results/discussions. For example: you mention the possibility of metadata corruption via overflowing into a free block. This very interesting consideration doesn't seem to ever be used/discussed. Same for the bit about page frame borders or adjacent objects of the same size but different type. (incidentally, isn't the same true for dlmalloc?)

 - clearer threat model: it's probably worth clarifying what the threat model is (early in the paper), the gap between the results that you will obtain/have obtained and the knowledge available to an adversary under your threat model. For example, as you only later point out, /proc/slabinfo is not world-readable and so it's probably worth clarifying earlier on that such information is not available to the adversary. It would also be interesting to clarify what kind of infoleak would be necessary to at least partially bridge this gap in a more practical scenario. For example, how what chances would an attacker have to prototype on a test system a special "reset" sequence followed by a sequence of allocations that lead to the desired heap configuration, and then reproduce this on a target system? I was hoping to read some of that in section 9.2, but it focused instead on stabilising results in a more synthetic scenario.

 - clearer "delta" with respect to Heelan's thesis. There's nothing wrong to build on top of existing research, but I think it would be very beneficial to clearly spell out what it is that your paper brings on top of the existing work.

 - deeper explanation of the used frameworks. For example, section 6 seems to require intimate knowledge of Heelan's work. For example, AppendFstSequence is mentioned there without any kind of introduction.

In what follows I have collected some minor points:

 - you mention heap feng shui, heap grooming or heap layout manipulation in section 3. Given that there are no space limits, could you say a few words about them and clarify how they are different?
 - in section 4.1, you say that "when this object is allocated again, there is no need to initialise it again". What exactly do you mean? Surely some initialisation is still necessary - could you please clarify?
 - in section 4.2 you refer to struct page assuming that the reader knows what it is/does. You might want to be a bit more clear in that regard.
 - in section 4.2 you mention "overflowing between page frame borders [42]"; it would be very interesting to spend more time on presenting what this is (and as I mentioned above, in further evaluating how this aspect influences the problem of heap layout manipulation).
 - how exactly do you introduce noise in the synthetic benchmarks (re section 7.1)?
 - let me preface this by saying that the paper is generally well written. While this is only a matter of personal taste, I would encourage you to use a more formal writing style. In particular, I would
    - always use the present tense
    - avoid open questions in the paper
    - avoid exclamations or colloquial expressions (easier said than done)
 - a few typos:
    - "which's"
    - "can not" (multiple times)
    - "exemplary"

(*) and in saying only I'm not discounting all the good work that went into building the system, I'm merely attempting to measure the delta in terms of contributions and lessons learned that the paper is offering to the community



**Expertise:**

Follow the literature closely, last published 5+ years ago

**Useful:**

yes

---

### Official Review · Reviewer_6dTG · 2022-06-03
**Linux kernel heap exploitation techniques and analysis**

**Decision:**

Weak accept: good paper with flaws that can be fixed in three months

**Review:**

Summary:

The paper focuses on Linux kernel heap layout manipulation and exploitation, and it brings forward a number of contributions: 1) the development of a framework to evaluate heap layout manipulation algorithms for the Linux kernel, 2) a performance benchmark to evaluate these algorithms, and 3) the proposal and evaluation of two algorithms, random search and KEvoHeap, based on genetic algorithms. The majority of these techniques were already published by Heelan et al., but they were focusing on userspace heap: the net contribution of this paper is to port these techniques to the Linux kernel, which is not trivial.

Pros:

- interesting work in an interesting research area
- paper is overall easy to read, and it contains a lot of background information
- the paper makes it very clear what it took from previous work and what are the "net" contributions
- it seems that (at least part of?) the work is / will be open sourced

Cons:
- the proposed approach, benchmark, and algorithms are all taken from existing works (but there are some tweaks that were necessary)
- the threat model is unclear (and even if one can guess, it should be spelled out)
- no concrete evaluation in realistic settings


Thanks the authors for this submission! I enjoyed reading this paper. I'm not a super expert on kernel heap exploitation, but I believe I was able to read and understand the paper without problems. I appreciate the authors took the time to discuss a number of background information and technical details before diving into the technical contribution of the paper.

The contributions are heavily based by existing work from Heelan et al., who were focusing on userspace heap attacks. The main contribution of this paper is to port the techniques/contributions by Heelan (framework + approach) from user-space heap layout manipulation/exploitation to the Linux kernel's. Even though the vast majority of the proposed techniques appear exactly the same, I really appreciate that the authors do not hide it and state the similarity (or the exact match) upfront; and, the authors provide a good discussion 1) explaining how porting these techniques from user space to kernel space was not trivial, and 2) explaining the rationale behind the difference. This makes the reader's job much easier. Thanks for that. Overall, I believe the difference are indeed not trivial, and the discussion and implemented differences are valuable. This work is also going to be open sourced (at least in part? see below), so it should be possible to build on top of it.

With that being said, I do have a number of concerns for this paper.

The first aspect is that the threat model is not clear, and even if one could guess it, I believe it should be properly spelled out. It also doesn't help that a reader familiar with kernel exploitation may assume a specific threat model... which then appears to be the wrong one. More in detail, when reading a paper about kernel exploitation, my assumption is that the threat model is "the attacker has arbitrary user-space code execution, and she wants to exploit a bug in the kernel to become root or achieve kernel-level code execution". Thus, I would have assumed that the attacker does not have access to root-level information or capabilities. But, when reading the paper, it becomes clear that the proposed system/mechanism does rely on "privileged information or mechanisms", such as accessing /proc/slabinfo, kprobes, or even having access to external QEMU resets. These mechanism are, of course, not accessible to an attacker who wants to perform local privilege escalation... and from this, I guess the authors are assuming a different threat model: not an attacker who wants to actually mount a LPE attack, but a vulnerability researcher who wants to show the exploitability of an heap-related bug that she found. Is my understanding correct? Regardless, I believe it's very important to be up front with the assumed threat model and which tools are available to the attacker/vuln researcher.

Another problem of the paper is that it is often not clear what are the implications of some "shortcuts" used to make the approach/discussion tractable. E.g., focusing on using a custom cache: this is mentioned in page 8, but the potential real-world applicability repercussions of this choice are discussed (in part) at the end of the paper, ~8/9/10 pages later. There are a number of similar decisions that simplify the implementation and evaluation of the proposed approach, but I believe it should be made crystal clear which decisions are "OK" and repercussion-free in terms of real-world applicability, and which are actually constituted a simplification that does make the discussion less realistic.

My biggest concern is about the applicability of this approach in the real-world, for what concerns a vuln researcher who wants to prove her real-world bug is exploitabile, and for what concern an attacker who would like to use this approach to be assisted for the exploitation. While there are many "stability guarantees" when one deals with userspace heaps, the situation is much messier (to the point of unexploitability without additional bugs?) when dealing with kernel-level heaps. While the paper presents an evaluation on the synthetic dataset (+ many assumptions!), the paper only has a discussion-level evaluation for what concerns using this system in a real-world scenario... and still, the real-world scenario considered is not the one of an attacker, but one in which the vuln researcher already has privileged access and "just" wants to show exploitability. What is the hope that such an approach could be used to assist in real-world exploitation? Under which assumptions (e.g., 'the heap is already defragmented', 'the initial state of the heap is know') would this tool be useful in real-world scenarios?

One comment on open sourcing: the paper mentions in a couple of points in the paper that "the code is on GitHub [18]". It is not clear whether everything about this paper is / will be on github, or just the two parts that were specifically in the paper? Would the authors mind clarifying whether the github repo contains all the artifacts? If that's the case, I suggest the authors to add a pointer about this aspect at the end of the introduction so that it's clear that everything will be available. It is a very important (positive) component, and I believe it's important to point this out!


Nitpick: the English is too informal at points. See usage of "e.g." (which, btw, should be "e.g.,") in the middle of sentences, or terms such as "huge", "skyrocketing", etc.


Some related works that the authors may want to consider adding, at least for context (nothing of this appears to be a critical omission, but these could be good to contextualize existing works in the area):
- HeapHopper: Bringing Bounded Model Checking to Heap Implementation Security, USENIX Security 2018
- Playing for K(H)eaps: Understanding and Improving Linux Kernel Exploit Reliability, USENIX Security 2022

**Expertise:**

Follow the literature closely, last published 5+ years ago

**Useful:**

yes

---

### Official Review · Reviewer_R9qg · 2022-06-14
**Kernel-SIEVE: Weak accept**

**Decision:**

Weak reject: interesting papers with flaws, not sure if they can be fixed in three months

**Review:**

### Summary
The authors propose a new framework (Kernel-SIEVE) for evaluating heap layout manipulation algorithms in kernel space. They implement and compare two algorithms: a naive pseudo-random search, and a more advanced genetic algorithm, KEvoHeap. The genetic algorithm the authors present performs well even in realistic conditions where multiple concurrent threads are making use of the kernel heap.

### Strenghts
- First attempt at evaluating kernel heap manipulation techniques:
The framework introduced by the authors is extensible and looks like it can be used to evaluate more techniques.

- In-depth explanation of the details:
The authors extensively describe both the algorithms, without excluding implementation details. This immensely helps reproducibility.
Furthermore, the authors provide detailed explanations and examples to justify the results of their evaluation. I also appreciated the effort the authors put into writing the background sections: they introduce the problem space well and guide the reader into the necessary concepts for understanding the rest of the paper.

- Insights in the porting of SIEVE to kernel space:
The real strength of this paper are the subtle fixes and clever workarounds that were necessary to port a heap manipulation framework to kernel space. In particular, I found the creation of a custom kernel cache for SLUBs a smart hack that I did not know was possible.

### Weaknesses

- Wrong assumption on CONFIG_SLAB_FREELIST_RANDOM:
In page 4, the authors state that the free list randomisation kernel feature is out of scope for this study, due to it being disabled in the Linux default configuration.
Unfortunately, while it is true that is disabled in the Linux default configuration, individual distributions often tweak and ship custom kernel configurations. I manually tested only Ubuntu and Arch Linux, but in both I found the CONFIG_SLAB_FREELIST_RANDOM to be enabled by default. I suspect that the vast majority of Linux distributions have this security feature enabled.
This is a major issue, as it would seriously undermine the applicability of the authors’ approach in real world scenarios. The authors state that there might be workarounds for this issue; they should do a deeper study on what they could be and if they will affect the KernelSIEVE’s evaluating abilities.

- Limited novelty:
Most of the work represents an extension of the work by Heelan and his SIEVE framework for userspace heap manipulation. This limits the scientific novelty to the insights required to port Heelan’s approach to kernel space. From my understanding, the only significant modifications to Heelan's approach are those required to make it work in the kernel (other than minor changes in the evolution selection of KEvoHeap). The introduction and evaluation of a new algorithm other than the two Heelan proposed would definitely improve this paper. Nevertheless, the authors’ extensively clarify which idea was taken from prior work and which idea instead was introduced in this paper, in a nice display of scientific correctness.

- Limited evaluation:
The current evaluation is not enough to convince a reader of the real-world applicability of their work. A table such as Table 2 is necessary for the QEMU evaluation, as Figure 18 alone is not enough (especially considering that some values were estimated instead of measured). If the QEMU reset is too big of a performance bottleneck, the authors should look into alternative solutions (an example might be state-of-the-art snapshot fuzzers such as Nyx, that achieve thousands of executions per second)

### Writing issues
While the paper was generally well-written, I spotted the following minor issues:
 - Looks like there is a misplaced e.g. in the first page (near “Zerodium”)
 - “Which’s” in the abstract
 - Page 8: no capitalisation after colon
 - Too much focus on technical details, too long paper:

Many parts of this paper (especially section 5 and 6) go into too much technical details of the implementation; I feel like they should best be placed in an appendix or in a README of the source code. So much information distracts the reader from the important design choices. I feel like the same concepts could be expressed in around 10-12 pages (instead of the current 18). Another example is Figure 6: the text describing the system is very clear and the figure does not add much, while still consuming almost half a page.

### Conclusion
While the paper is an extension of a prior technique, and the proposed approach has limited applicability in the real world, I still think that some of the insights presented here are interesting and should be shared with the rest of the scientific community.
For this reason, I will push for a weak accept of this paper.
It is vital though that the authors address the randomisation of freelists, and do a more through study of applicability in the real world.

**Expertise:**

Follow the literature closely, last published 5+ years ago

**Useful:**

yes

---

### Meta-Review · Area_Chair_ucbJ · 2022-06-15

**Recommendation:** Revise
**Confidence:** 4

**Metareview:**

Thank you for submitting to JSys. This is an interesting paper that spurred good discussions among the reviewers.
The authors propose a new heap layout manipulation engine for the Linux kernel. While similar approaches have been applied to user space, the kernel allocators remain somewhat opaque. While kernel allocators are often simpler than user space (i.e., user space allocators have arenas, per thread pools, and other optimizations), there are many more kernel allocators alive at the same time.
The presented kernel-SIEVE framework automates heap manipulation for the Linux kernel along with an evaluation of two approaches (a pseudo-random approach and a genetic algorithm).
All three reviewers agree that the approach is interesting and targets a challenging problem. The paper is well formatted and presents a good structure that introduces the key concepts well.
While the reviewers appreciated the above mentioned key points, there were also several issues that require attention.
First, the threat model is not clear. I.e., the power of the attacker is somewhat opaque and needs to be clarified (it should be a researcher with a known bug aiming for an exploit).
Second, the authors should demonstrate the feasibility of exploitation by providing an end to end example.
Third, the coupling should be expanded from kmalloc to system calls. Instead of tying the exploit primitives to simple kmallocs in the code that may (or may not) be under the control of the attacker, it should be adjusted to start from system calls.
Fourth, metadata overflowing into other blocks is not discussed (but should be).
Fifth, related work is handled incompletely. Especially Heelan's thesis should be discussed in much more detail. As the paper heavily builds on these abstractions (that were demonstrated by Heelan in user space), the authors need to justify the additional complexity for kernel space.
In short, the reviewers think that there is merit but the authors need to expand the evaluation with this end to end example and an abstraction that ties to system calls instead of kmallocs. This additional evaluation and slight design change calls for a major revision.
The authors have agreed on the following criteria:

---

The paper expands on the thesis from Heelan and moves the attacks to kernel space. The main contribution is the porting of this technique and abstracting them towards a new environment.

* Clarify the threat model
* Provide an end to end example on how this work orchestrates an attack from system call to the final compromise.
* Expand the coupling with the Linux kernel and abstract from kmalloc's to system calls (or provide sufficient reasoning that kmallocs are easily reachable in sufficient cases)
* Discuss the metadata overflowing into a free block with an example and provide more details
* Discuss Heelan's thesis in more detail (and justify the delta). Relying on other works is fine but needs to be discussed
* Add missing related work

Further, the open source strategy is not clear, especially how much of the paper will be opensourced. The authors should discuss this in a comment.

---

---

### Meta-Review · Area_Chair_ucbJ · 2022-12-15

**Recommendation:** Accept (with shepherding)
**Confidence:** 4

**Metareview:**

The authors provided an updated paper where they addressed large parts of the major revision requirements. The authors integrated a discussion of an extended example along with updating the threat model and better introduction of the technique in the abstract/design sections of the paper. Those parts were well handled.

Generally, the reviewers felt that the revision was not complete and that some more tuning is needed. We therefore suggest a minor revision/shepherding to more effectively work out the final details.

In general, the reviewers would like to see:

clarification on the threat model on where privileged information is necessary (or demonstrating using a set of real examples that it is not necessary); while the threat model has been updated, it remains vague regarding this extra privileged information. This should be clarified for the final version.

The real-world-attack is more of a case study based on a vulnerable kernel module and should be phrases like that (or replaced with a real example)

If you can, provide an end to end example on how this work orchestrates an attack from system call to final compromise using a REAL vulnerability.

The first two points are strongly suggested, the third point is optional. The reviewers agreed that the authors satisfied most of the points and recommend a minor revision with a shepherd to work through the final three points (where two of them are mandatory, one of them is optional).

---

### Decision · Program_Chairs · 2022-06-15

**Decision:**

Accept (with shepherding)

**Comment:**

Decision updated: Accept (with shepherding)
====
Dear authors,

Thank you for submitting to JSys.

The reviewers and the area chairs have agreed on a "Revise" decision for your manuscript. As per JSys policy, you have up to 3 months to address the points listed in the Meta-review (see below) and resubmit your manuscript. If those points are satisfactorily addressed, the manuscript will be accepted for publication. The full reviews will be made available shortly, we apologize for the slight delay.

When submitting your revised manuscript, please highlight the main updates in the paper to facilitate the second round of reviewing.

Also note that, as per JSys policy, all solution papers must pass the Artifact Evaluation [1] before being published. Unless I'm mistaken, there has been no artifact submitted thus far related to this paper. We encourage you to submit your artifact for the next deadline (Aug. 1st) to avoid delaying the publication of your manuscript after the second round of review.

We look forward to receiving your revision.

[1] https://www.jsys.org/cfp/#artifact-evaluation